# General agents need world models

Jonathan Richens [1]   Tom Everitt [1]   David Abel [1]

## Abstract

Are world models a necessary ingredient for flexible, goal-directed behaviour, or is model-free learning sufficient? We provide a formal answer to this question, showing that any agent capable of generalizing to multi-step goal-directed tasks must have learned a predictive model of its environment. We show that this model can be extracted from the agent's policy, and that increasing the agents performance or the complexity of the goals it can achieve requires learning increasingly accurate world models. This has a number of consequences: from developing safe and general agents, to bounding agent capabilities in complex environments, and providing new algorithms for eliciting world models from agents.

## 1. Introduction

A hallmark of human intelligence is the ability to perform novel tasks with minimal supervision, formalised by few-shot and zero-shot learning (Lake et al., 2017). With the emergence of these capabilities in language models (Brown et al., 2020), focus has shifted to developing general agents—systems capable of performing long horizon goal-oriented tasks in complex, real-world environments (Yao et al., 2022; Hao et al., 2023). In humans this kind of flexible goal-directed behaviour relies heavily on rich mental representations of the world, i.e. world models (Johnson-Laird, 1983; Ha & Schmidhuber, 2018), which are used to set abstract goals beyond immediate sensory inputs (Locke & Latham, 2013), and to deliberatively and proactively plan actions (Bratman, 1987). Whether world models are necessary for achieving human-level AI has long been debated, pitting the challenges of learning models against the potential benefits they confer (Huang, 2020).

Explicitly model-based agents have achieved impressive performance across many tasks and domains (Hafner et al.,

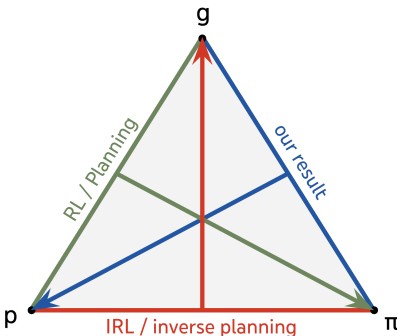

Figure 1: Our result complements previous insights from planning and inverse RL. While planning uses a world model and a goal to determine a policy, and IRL and inverse planning use an agent's policy and a world model to identify its goal, our result uses an agent's policy and its goal to identify a world model

2023; Wang et al., 2023; LeCun, 2022; Raad et al., 2024), and having direct access to the agent's world model has benefits like being able to apply formal planning methods (Sutton, 2018), predicting the agent's behaviour in safety-critical domains (Amodei et al., 2016; Dalrymple et al., 2024), reducing sample complexity (Hafner et al., 2019) and supporting transfer learning (Chua et al., 2018; Zhu et al., 2023). However, learning accurate models of real-world systems can be extremely challenging (Dulac-Arnold et al., 2019), and the performance of model-based agents is fundamentally limited by their model's fidelity.

In "Intelligence without representation", Brooks famously proposed that *the world is its own best model*, and that all intelligent behaviours can emerge in model-free agents interacting through action-perception loops, without needing to learn explicit representations of the world (Brooks, 1991). This view has largely been borne out by the development of model-free agents capable of generalizing across a wide range of tasks and environments (Reed et al., 2022; Brohan et al., 2023; Driess et al., 2023; Black et al., 2024; Schrittwieser et al., 2020). This model-free paradigm aims to achieve truly general agents while side-stepping the challenges inherent in learning a world model. However, there is mounting evidence that model-free agents may in fact learn *implicit* world models (Li et al., 2022), and may even

[1]Google DeepMind. Correspondence to: Jonathan Richens <jonrichens@google.com>.

*Proceedings of the 42nd International Conference on Machine Learning*, Vancouver, Canada. PMLR 267, 2025. Copyright 2025 by the author(s).

learn implicit planning algorithms (Hou et al., 2023; Bush et al., 2025).

This raises a fundamental question: is there a model-free shortcut to human-level AI? Or is learning a world model necessary, with all the complexity this entails? And if so, just how accurate and comprehensive do world models need to be to support a given level of capability? We provide a formal answer to these questions, showing that,

> *any agent that satisfies a regret bound for a sufficiently diverse set of simple goal-directed tasks must have learned an accurate predictive model of its environment.*

Specifically, we consider environments described by a fully observed Markov processes, and propose a minimalist definition of general agents as goal-conditioned policies (Liu et al., 2022) that satisfy a regret bound for a large set of simple goal-directed tasks (such as steering the environment into a desired state). We derive an algorithm that returns an approximation of the environment transition function (a world model) given the policy of any such agent, and show that the error in this approximation decreases as we increase the agent's performance or the complexity of the goals it can achieve. In other words, general agents *are* world models, with all the information required to simulate the environment encoded in their policy. Importantly, we prove this for any agent that satisfies a regret bound, regardless of the details of its training and architecture and without imposing rationality assumptions.

The necessity of learning a world model has profound consequences for how we develop general AI systems, how capable these systems can ultimately be, and how we can ensure agents are safe and interpretable. We explore these consequences and others in Section 4. A more immediate consequence is that in proving our result we derive new algorithms for extracting world models from general agents. We demonstrate this in Section 3.1, and show that our algorithms can recover accurate world models even when the agent strongly violates our competence assumptions. In Section 5 we then discuss related work, including inverse reinforcement learning and mechanistic interpretability.

## 2. Setup

### 2.1. Notation

Capital letters denote random variables $X$ and lower case letters $x$ denoting a value or state $X = x$. Bold letters denote sets of variables $\boldsymbol{X} = \{X_1, X_2, \ldots, X_m\}$ and $\boldsymbol{x}$ denotes the joint state $\{x_1, x_2, \ldots, x_m\}$. Square brackets denote a proposition, e.g. $[X = x]$ is True if $X = x$ and False otherwise.

### 2.2. Environment

We assume the environment is a controlled Markov process (cMP) (Puterman, 2014; Sutton, 2018), which is a Markov decision process without a specified reward function or discount factor. Formally, a cMP consists of a set of states $\boldsymbol{S}$, a set of actions $\boldsymbol{A}$, and a transition function $P_{ss'}(a) = P(S = s' \mid A = a, S = s)$. We refer to a sequence of state–action pairs over time as a *trajectory*, $\tau = (s_0, a_0, s_1, a_1, \ldots)$ and a finite prefix of $\tau$ as a *history*, $h_t = (s_0, a_0, \ldots, s_t)$.

**Definition 1** (Controlled Markov process). *A controlled Markov process (cMP) is a Markov decision process (MDP) without a specified reward function or discount factor. It is defined by the tuple $(\boldsymbol{S}, \boldsymbol{A}, P_{ss'}(a))$ where $\boldsymbol{S}$ is the state space, $\boldsymbol{A}$ is the action space, and $P_{ss'}(a) = P(S = s' \mid A = a, S = s)$ is the transition function.*

To derive our results we make the standard assumptions that the environment is finite, irreducible, and stationary, meaning every state is reachable from every other state under some finite sequence of actions, and transition probabilities do not change over time. Furthermore we assume $|\boldsymbol{A}| \geq 2$ so that the environment can support non-trivial policies.

**Assumption 1.** *We assume the environment is described by an irreducible, stationary, finite, controlled Markov process (Def. 1) with at least two actions.*

For further discussion of these standard assumptions see Puterman, 2014; Sutton, 2018.

### 2.3. Goals

Our aim is not to provide a complete definition of goal-directed behaviour, but to define a simple and intuitive class of goals we might reasonably expect an agent to be capable of. In many settings including planning (Ghallab et al., 2004), goal-conditioned reinforcement learning (Liu et al., 2022), and control theory (Åström & Murray, 2021)), the simplest goals are desirable states of the world (goal states), and a goal is achieved by the agent steering the environment into one of these goal states. More generally, goal-directed behaviour can involve a sequence of sub-goals to be achieved in a particular order, and may include desirable actions as well as environment states. This class includes instruction following, which is the type of goal-directed behaviour we typically desire of AI agents.

To describe these sequences of sub-goals (sequential goals) we use Linear Temporal Logic (LTL) (Pnueli, 1977; Baier & Katoen, 2008), which is commonly used to specify tasks and temporal objectives for agents (Littman et al., 2017; Li et al., 2017; Hasanbeig et al., 2019; Dzifcak et al., 2009; Ding et al., 2014) including more recently for goal-conditioned reinforcement learning agents (Vaezipoor et al., 2021; Qiu

et al., 2023; Jackermeier & Abate, 2024). An LTL expression $\varphi$ assigns a truth value to each trajectory (denoted $\tau \models \varphi$), which is true iff $\tau$ satisfies the LTL expression. Concretely, we define a goal as pair $(\mathcal{O}, \boldsymbol{g})$ where $\boldsymbol{g}$ is a set of goal states and $\mathcal{O}$ is a temporal operator specifying a time horizon within which the goal states should be reached. For our results it will be sufficient to restrict our attention to two temporal operators; Eventually ($\diamondsuit$), where the goal state must be reached at any future time, and Next ($\bigcirc$), where the next state must be a goal state, e.g. to capture the immediate consequences of an agent's actions. In the absence of a temporal operator, the goal condition must in the current time step, which we refer to as Now and represent with the trivial (True) operator $\top$. We denote goals as $\varphi := \mathcal{O}([(s, a) \in \boldsymbol{g}])$. For example, $\varphi = \diamondsuit([S = s])$ specifies that state $s$ must eventually be reached. See Appendix A.3 for further discussion.

**Definition 2** (Goals). *A goal $\varphi$ is an LTL expression of the form $\varphi = \mathcal{O}([(s, a) \in \boldsymbol{g}])$ where,*

- *$\boldsymbol{g}$ is a set of goal-states, a sub-set of the joint states of the environment-agent system $(s, a) \in \boldsymbol{S} \times \boldsymbol{A}$,*

- *$\mathcal{O}$ is a temporal operator specifying the time horizon for reaching $\boldsymbol{g}$. We restrict to $\mathcal{O} \in \{\bigcirc, \diamondsuit, \top\}$ where $\bigcirc$ = Next, $\diamondsuit$ = Eventually, $\top$ = Now.*

Using Def. 2 we can construct composite goals of increasing complexity by either combining goals in sequence (where goal $\varphi_A$ must be achieved before goal $\varphi_B$) or in parallel (where satisfying either goal $\varphi_A$ or goal $\varphi_B$ is sufficient). We use $\psi = \langle \varphi_1, \ldots, \varphi_n \rangle$ to denote a sequence of sub-goals, where the agent must satisfy $\varphi_1$ before moving on to $\varphi_2$, and so on. Here $\psi$ is also an LTL expression, which we provide a formula for in Appendix A.3. We refer to $n$ as the *depth* of $\psi$, i.e. the number of sub-goals the agent must satisfy to satisfy $\psi$ (also known as the temporal height, Demri & Schnoebelen (2002)). Parallel composition is achieved by taking the disjunction (OR) of two or more (sequential) goals, i.e. for $\psi' = \psi_1 \vee \psi_2$, $\tau \models \psi'$ is true iff $\psi_1$ or $\psi_2$ are satisfied by $\tau$. Finally, $\boldsymbol{\Psi}$ denotes the set of all composite goals for a given environment, and $\boldsymbol{\Psi}_n$ to denote the set of all compositions of goals (Def. 3) of depth at most $n$.

**Definition 3** (Composite goals). *A sequential goal $\psi$ is an ordered sequence of sub-goals (Def. 2) $\psi = \langle \varphi_1, \ldots \varphi_n \rangle$, where the agent must achieve sub-goal $\varphi_i$ before $\varphi_{i+1}$. The depth of a sequential goal is the number of sub-goals $depth(\psi) = n$. A composite goal is a disjunction of one or more sequential goals $\psi = \bigvee_{i=1}^{m} \psi_i$, i.e. the agent must achieve any sub-goal $\psi_i$ to achieve $\psi$. The depth of a composite goal is the max depth of its sub-goals $depth(\psi) = \max_{\psi_i} depth(\psi_i)$. $\boldsymbol{\Psi}_n$ is the set of all composite goals $\psi$ with $depth(\psi) \leq n$.*

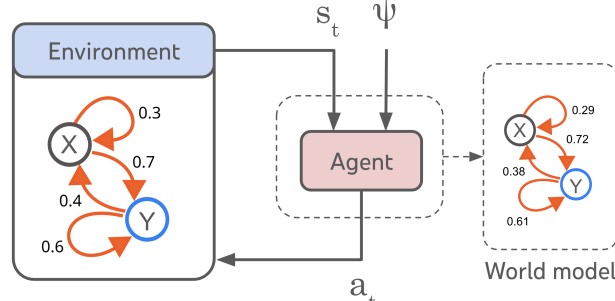

Figure 2: The agent-environment system. Agents are maps from states $s_t$ (or histories) and goals $\psi$ to actions $a_t$. The dashed line represents Algorithm 1, which recovers the environment transition probabilities from this agent map.

**Example:** A maintenance robot is given the task of fixing a faulty machine, or finding an engineer and alerting them that the machine is broken. Fixing the machine requires performing a sequence of predetermined actions $a_1, a_2, \ldots, a_N$ and each time attaining the desired outcome $s_1, s_2, \ldots, s_N$, which can be represented as the sequential goal $\psi_1 = \langle \varphi_1, \varphi_2, \ldots, \varphi_N \rangle = [A = a_1, S = s_1] \wedge \bigcirc([A = a_1, S = s_1] \wedge \bigcirc(\ldots))$ (using simplified notation for $[(s, a) \in \boldsymbol{g}]$). Finding and alerting the engineer requires the robot to navigate to an engineer $S = s_{\text{eng}}$ and alerting them $A = a'$, $\psi_2 = \diamondsuit([S = s_{\text{eng}}, A = a'])$. The robot's goal can be represented as the composite goal $\psi = \psi_1 \vee \psi_2$.

### 2.4. Agents

Our aim is to formulate a minimalist definition of an agent capable of achieving a range of goals in its environment. To this end we focus on goal-conditioned agents (Liu et al., 2022; Schaul et al., 2015), which are policies $\pi$ that map histories and goals to actions, $\pi : h_t, \psi \mapsto a_t$ (Figure 2). Note this does not restrict us to agents that can condition their actions on the full history of the environment, as any policy (e.g. a Markov policy) can be represented in this way. For simplicity, we assume that the environment is fully observed by the agent, and that the agent follows a deterministic policy. This leads to a natural definition of an optimal goal-conditioned agent for a given environment and set of goals $\boldsymbol{\Psi}$, which is a policy that maximizes the probability that $\psi$ is achieved, for all $\psi \in \boldsymbol{\Psi}$.

**Definition 4** (optimal goal-conditioned agent). *For a given set of goals $\boldsymbol{\Psi}$ (Def. 3) an optimal agent is a goal-conditioned policy $\pi^*(a_t \mid h_t; \psi)$ where $\pi^*$ is deterministic and satisfies,*

$$\pi^* = \arg\max_{\pi} P(\tau \models \psi \mid \pi, s_0) \qquad (1)$$

*$\forall s_0$ s.t. $P(s_0) > 0$, where $s_0$ is the initial state of the*

*environment at $t = 0$, and $\forall \, \psi \in \Psi$.*

Real agents are rarely optimal, especially when operating in complex environments and for tasks that require coordinating many sub-goals over long time horizons. Hence, we relax Def. 4 to define a *bounded* agent that is capable of achieving goals of some maximum goal depth $\Psi_n$ with a failure rate that is bounded relative to the optimal agent. Bounded agents are defined by two parameters; i) a *failure rate* $\delta \in [0, 1]$, which places a lower bound on the probability that the agent achieves a goal compared an optimal agent (analogous to regret), and ii) a maximum goal depth $n$, such that this regret bound holds only for goals with a depth less than or equal to $n$. This naturally captures the type of agents we are interested in—those which have some capability (parameterised by $\delta$) for achieving goals of some maximum complexity $\Psi_n$.

**Definition 5** (bounded goal-conditioned agent). *A bounded goal-conditioned agent is a goal-conditioned policy $\pi(a_t \mid h_t; \psi)$ satisfying,*

$$P(\tau \models \psi \mid \pi, s_0) \geq \max_{\pi} P(\tau \models \psi \mid \pi, s_0)(1 - \delta) \quad (2)$$

*$\forall \, \psi \in \Psi_n$ where $n$ is the maximum goal depth and $s_0$ is the initial state of the environment at $t = 0$.*

Importantly, Def. 5 only assumes a level of *competence* for the agent. We do not, for example, impose any rationality assumptions on the agent as in (Von Neumann & Morgenstern, 2007; Savage, 1972), which are not satisfied by current agents (Raman et al., 2024b).

**Example:** Following the previous example, the performance of the maintenance robot is measured by the probability that it either fixes the machine or alerts an engineer, i.e. $P(\tau \models \varphi_1 \vee \varphi_2 \mid \pi, s_0)$. This intuitively involves weighing up the two possible courses of action; if the repair is difficult then directly attempting it could lead to failure, and finding an engineer is the better course of action. Or if the probability of finding an engineer is very low, attempting to fix the machine may be the best strategy. Whatever the agent chooses to do, we can measure its performance relative to the probability that the optimal agent will solve the task, $P(\tau \models \varphi_1 \vee \varphi_2 \mid \pi^*, s_0)$.

## 2.5. World models

We are interested in the role of world models in goal-directed behaviour. Hence we focus on predictive world models, which can be used by agents to plan. This follows the definition of world models used in reinforcement learning (RL), as opposed to the use of the term to describe representations of the environment state alone (e.g. in Li et al. (2022); Gurnee & Tegmark (2023b)). For model-based RL agents, explicit world models are usually one-step predictors of the environment state (Sutton, 2018), which in Markovian environments are sufficient to predict the evolution of the environment under arbitrary policies. We define a world model as any approximation $\hat{P}_{ss'}(a)$ of the transition function of the environment (Def. 1) $P_{ss'}(a) = P(S_{t+1} = s' \mid A_t = a, S_t = s)$, with bounded error $|\hat{P}_{ss'}(a) - P_{ss'}(a)| \leq \epsilon$.

## 3. Results

Our main result is a proof by reduction—we assume the agent is a bounded goal-conditioned agent (Def. 5), i.e. it has some (lower bounded) competency at goal-directed tasks of some finite depth $n$ (Def. 3). We then prove that an approximation of the environment's transition function (a world model) is determined by the agent's policy alone, with bounded error. Hence, learning such a goal-conditioned policy is informationally equivalent to learning an accurate world model.

**Theorem 1.** *Let $P_{ss'}(a) = P(S_{t+1} = s' \mid A_t = a, S_t = s)$ be the transition probabilities of an environment satisfying Assumption 1. Let $\pi$ be a goal-conditioned agent (Def. 5) with a maximum failure rate $\delta$ for all goals $\psi \in \Psi_n$ where $\Psi_n$ is the set of all composite goals with maximum goal depth $n > 1$. $\pi$ fully determines a model for the environment transition probabilities $\hat{P}_{ss'}(a)$ with errors satisfying*

$$\left| \hat{P}_{ss'}(a) - P_{ss'}(a) \right| \leq \sqrt{\frac{2P_{ss'}(a)(1 - P_{ss'}(a))}{(n-1)(1-\delta)}}$$

*for any $n, \delta$, and for $\delta \ll 1$, $n \gg 1$ the error scales as,*

$$\left| \hat{P}_{ss'}(a) - P_{ss'}(a) \right| \sim \mathcal{O}\left( \delta/\sqrt{n} \right) + \mathcal{O}(1/n)$$

*Proof in Appendix A.6.*

In Appendix A.5 we give a simplified overview of the proof of Theorem 1. We derive an algorithm that queries the goal-conditioned policy with different goals $\psi \in \Psi_n$ which correspond to either-or decisions between two incompatible sub-goals $\psi = \psi_a \vee \psi_b$. As the agent satisfies a regret bound, its choice of action encodes information about which of the sub-goals has a higher maximum probability of being satisfied, and this information can be used to estimate the transition probabilities $\hat{P}_{ss'}(a)$. We then prove that this estimate satisfies the error bounds stated in Theorem 1. Note that while the statement of Theorem 1 assumes the agent has a maximum failure rate (regret bound) $\delta$ for all $\psi \in \Psi_n$, in fact our proof only requires the agent satisfies this regret bound for a small subset of $\Psi_n$ consisting of $n$ composite goals (see discussion of emergent capabilities in Section 4).

Our algorithm for recovering a bounded-error world model from a bounded goal-conditioned agent (Algorithm 1) is detailed in Appendix C. It is universal, meaning the same

algorithm works for all agents satisfying Def. 5 and all environments satisfying Assumption 1. It is also unsupervised; the only input to the algorithm is the agent's policy $\pi$. The existence of this algorithm, which converts $\pi$ into a bounded error world model, implies the world model is encoded in the agent's policy, and learning such policy is informationally equivalent to learning a world model. Formally, the approximate world model $\hat{P}_{ss'}(a)$ is *identifiable* given the agent's policy and our assumptions (see for example Bareinboim et al. (2022)). In Section 5 we compare Algorithm 1 and its assumptions to methods for recovering world models in mechanistic interpretability, which similarly use the existence of a recovery map establish that an agent has learned a world model.

**Properties of the world model.** The accuracy of the world model recovered from the agent in Theorem 1 increases as the agent approaches optimality ($\delta \to 0$), and/or as the depth $n$ of sequential goals it can achieve increases. A key consequence of the derived error bounds is that for any $\delta < 1$ we can recover an arbitrarily accurate world model if we can make $n$ sufficiently large. Therefore, in order to achieve long horizon goals even with a high failure rate $\delta \sim 1$, the agent must have learned a highly accurate world model. The error bounds also depend on the transition probabilities, and dividing both sides of the bound by $P_{ss'}(a)$ shows that the relative error $\hat{P}_{ss'}(a)/P_{ss'}(a)$ can become very large for $P_{ss'}(a) \ll 1$. This means that for any $\delta > 0$ and/or finite $n$, there can exist low probability transitions that the agent is not required to learn. This matches the intuition that sub-optimal or finite-horizon agents need only learn relatively sparse world models covering the more common transitions, but achieving goals with higher success rate or longer horizons requires higher resolution world models.

Theorem 1 imparts only a trivial error bound on the world model we can extract from agents whose maximum goal depth is $n = 1$. It is not immediately clear if this means that agents that only optimize for immediate outcomes (*myopic agents*) do not need to learn a world model, or if Theorem 1 simply fails to capture this class of agents. To resolve this we derive a result for myopic agents, which satisfy a regret bound for $n = 1$ and only a trivial regret bound ($\delta = 1$) for any $n > 1$.

**Theorem 2.** *Let the set of myopic goals $\Psi_{myopic}$ be the subset of depth-1 composite goals $\Psi_1$ such that the goal state(s) must be attained immediately after the agents first action, $\varphi = \bigcirc[(s, a) \in g]$. We define an optimal myopic agent as a policy $\pi^*(a_t \mid h_t, \psi)$ that is optimal for all $\psi \in \Psi_{myopic}$. For an environment satisfying Assumption 1, any bounds on the transition probabilities $|\hat{P}_{ss'}(a) - P_{ss'}(a)| \leq \epsilon$ than can be determined from $\pi^*$ are trivial ($\epsilon = 1$) and tight. Proof in Appendix B.*

Theorem 2 implies that there exists no procedure that can

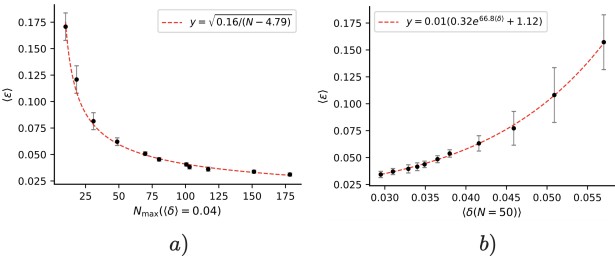

Figure 3: a) shows the mean error in the world model recovered by Algorithm 2, $\langle \epsilon \rangle$, decrease as the agent learns to generalize to higher depth goals. $N_{\max}(\langle \delta \rangle = 0.04)$ is the maximum goal depth such that the agent achieves a mean regret $\leq 0.04$. The scaling is $\mathcal{O}(n^{-1/2})$, as with the scaling between the worst-case error $\epsilon$ and worst-case regret $\delta$ in Theorem 1. b) shows the mean error scaling with $\langle \delta(n = 50) \rangle$, the mean regret the agent achieves for depth $n = 50$ goals. For both figures, error bars show 95% confidence intervals for the mean over 10 experiments where we re-trained the agents with different experience trajectories of the same length.

even partially determine the transition probabilities from the policy of a myopic agent. In the proof of Theorem 2 we show this by explicitly construct a myopic agent that is optimal for any choice of $P_{ss'}(a) \in [0, 1]$, and so the policy of such an agent can only impart trivial bounds on the transition probabilities. Therefore, learning a world model with is not necessary for myopic agents—world models only become necessary for agents pursuing goals with multiple sub-goals and over multi-step horizons.

### 3.1. Experiments

We demonstrate our procedure for recovering a world model from an agent, and how the accuracy of the model increases as the agent learns to generalize to more tasks (longer horizon goals). We also investigate if our algorithm can recover the transition function when the agent strongly violates our assumptions (Def. 5). Specifically, a realistic agent could be highly competent ($\delta \sim 0$) for some depth-$n$ goals but completely fail for others ($\delta = 1$). This agent would violate any non-trivial regret bound as in Def. 5, resulting in trivial model-error bounds in Theorem 1. To explore this case we relax Def. 5 and consider agents where the regret bound holds only on average over some set of goals $\Psi$, i.e. $\langle \delta \rangle \leq k$ where $\langle \delta \rangle$ is the average value of $1 - P(\tau \models \psi \mid \pi, s_0)/\max_\pi P(\tau \models \psi \mid \pi, s_0)$ over all $\psi \in \Psi$. We then determine empirically how the average error $\langle \epsilon \rangle$ in the world model recovered by Algorithm 1 scales with the agent's average regret $\langle \delta \rangle$ (Figure 3 b)), where $\epsilon := |\hat{P}_{ss'}(a) - P_{ss'}(a)|$ and $\langle \epsilon \rangle$ is mean value of $\epsilon$ over all transitions $(s, a, s')$.

The environment used to test our algorithms is a randomly generated cMP satisfying Assumption 1, comprising of 20 states and 5 actions with a sparse transition function. We train our agent using trajectories sampled from the environment under a random policy, and we increase the competency of our agent by increasing the length of the trajectory it is trained on, $N_{\text{samples}}$. See Appendix D for further details on the agent and experimental setup. We recover the world model using Algorithm 2, a simplified version of Algorithm 1.

As we increase $N_{\text{samples}}$ we observe the agent can generalize to longer horizon goals, captured by $N(\langle \delta \rangle = k)$ which is the maximum goal depth $n$ such that the agent achieves an average regret $\langle \delta \rangle = k$ for goals of depth $n$. We find that for all $N_{\text{samples}}$ tested, and for all goal deptsh $n$, our agent agent achieved a worst-case regret $\delta = 1$ for some goals, i.e. the agent violates any non-trivial regret bound of the form Def. 5. Nevertheless, we find that Algorithm 2 recovers the transition function with a low average error (Figure 3 b)), which scales as $\sim \mathcal{O}(n^{-1/2})$, like the error bound in Theorem 1. Hence, in spite of the agent violating our assumptions and achieving maximal regret for some goals, the average error has a similar decay with the goal depth as when the worst-case regret bound (Def. 5) is satisfied. Therefore, we can still accurately recover the transition function from the agent as long as it achieves a relatively low average regret for long horizon goals.

## 4. Discussion

We now discuss the consequences of Theorem 1 and its limitations.

**No model-free path to general agents.** Theorem 1 implies that any agent that satisfies a regret bound as in Def. 5 must have learned an implicit world model, and the accuracy of the model increases as the regret $\delta$ decreases or the maximum goal depth $n$ increases. In other words, there is no way to train an agent capable of generalizing to long horizon tasks without learning a world model, and the fidelity of the model bounds the agent's capabilities. This removes a key motivation for model-free approaches, as learning a world model cannot be avoided. On the other hand, it motivates explicitly model-based architectures (LeCun, 2022; Hafner et al., 2023; Schrittwieser et al., 2020), which can directly attack the model learning problem, and can exploit their benefits in terms of sample efficiency (Hafner et al., 2019), planning (Sutton, 2018), interpretability (Glanois et al., 2024) and safety (Amodei et al., 2016).

**Emergent capabilities.** An accurate world model is a powerful tool—it can be used to determine low-regret policies for *any* well-defined objective, without requiring further interaction with the environment or task-specific data. Hence,

implicit world models have been proposed as an explanation for emergent capabilities in foundation models (Brown et al., 2020; Li et al., 2022; Abdou et al., 2021). Our results support this hypothesis by revealing a mechanism by which implicit world models could emerge during training. To minimize regret across a variety of training tasks, agents are required to learn an implicit world model, which in turn could support generalization to a wide range of tasks the agent was never explicitly trained on. Note that for simplicity we have stated Theorem 1 with the assumption that the agent can generalize to any depth-$n$ composite goal $\mathbf{\Psi}_n$, but this is not the strongest statement of the result. In the proof (Appendix A.6) the agent is required to generalize only to a small subset of $\mathbf{\Psi}_n$, comprising of $n$ simple composite goals (see also Algorithm 1). There are likely many such choices of subsets of $\mathbf{\Psi}_n$ (e.g. a different sufficient set is used in Algorithm 2, Appendix C), and there are likely other tasks beyond achieving composite goals (Def. 3) that are sufficient to derive the result. Our findings therefore point to the existence of sets of simple tasks, where learning to perform these tasks implies sufficient world knowledge to (in principle) generalize to any task.

Beyond planning, world models support domain adaptation (Chua et al., 2018), reasoning about uncertainty (Lockwood & Si, 2022) and social cognition (Rabinowitz et al., 2018). With additional structural assumptions, they can also support causal reasoning (Pearl, 2018), simulating counterfactual trajectories and imagination (Racanière et al., 2017), and reasoning about intent (Ward et al., 2024) and attribution (Chockler & Halpern, 2004). Theorem 1 provides a simple explanation for how this wide range of cognitive abilities, associated primarily with human-level intelligence (Tomasello, 2022), can emerge from simple goal-directed behaviour. This could explain away several prominent theories for how these capabilities arose in nature, which propose specific environmental factors such as resource uncertainty (Hills et al., 2015) and social complexity (Dunbar, 1998) as the driving force for their emergence. The composite goals used in the proof of Theorem 1 describe simple either-or navigation tasks in a single-agent environment. If an agent was required to solve these tasks without repeated attempts (zero-shot), perhaps due to risk of death, this would require the agent to satisfy a regret bound as in Def. 5, and hence learn a world model capable of supporting these capabilities, without needing to invoke novel environmental or social factors.

**Safety.** Several proposals for AI safety and alignment require an accurate predictive model of the agent-environment system to verify the safety of plans (Bengio et al., 2024; Dalrymple et al., 2024), safely explore (Brunke et al., 2022), predict human responses (Leike et al., 2018), avoid problematic incentives (Farquhar et al., 2022), and incorporate model-based concepts into decision making such as intent

(Ward et al., 2024), deception (Ward et al., 2023) and harm (Richens et al., 2022; Bengio et al., 2024). Other proposals focus on passive oracles (essentially world models), avoiding agents altogether due to their inherent safety issues (Bengio et al., 2025; Armstrong & O'Rorke, 2017).

One major impediment to these approaches is the reasonable expectation that the capabilities of model-free agents will outpace our ability to learn accurate predictive models of complex real-world environments. There are already several examples of AI systems that can solve prediction tasks in domains we cannot yet model (Abramson et al., 2024; Merchant et al., 2023), and it is intuitively hard to interpret, audit and correct the behaviour of black-box agents operating in environments we do not understand, or where the agent has superior world knowledge than the supervisor (Christiano et al., 2021). Our results point to solution, providing a theoretical guarantee that we can extract an accurate world model from any sufficiently capable model-free agent. Importantly, the fidelity of this model increases with the agent's capabilities, especially as agent gets better at achieving goals over long time horizons—precisely the regime where safety concerns such as reward hacking become important (Farquhar et al., 2025). Future work should explore developing scalable algorithms for eliciting these world models and using them to improve agent safety.

**Limits on strong AI.** Our ability to learn accurate models of the world is fundamentally limited by the openness of real-world systems, their complexity and unpredictability, confounding, limited data, and the curse of dimensionality (Box & Draper, 1987). Theorem 1 implies that training an agent capable of generalizing to a wide range of tasks in the real world is extremely hard—at least as hard (and possibly much harder) than learning an accurate model of the world. While 'thinking slow' (Kahneman, 2011) deliberative planning and reasoning is not necessary (or even desirable) in every situation—for example, humans also generalize to novel tasks through 'thinking fast' heuristics (Tversky & Kahneman, 1974) and similarity-based generalization (Shepard, 1987)—our results establish that for any agent, natural or artificial, their ability to generalize is ultimately bounded by their ability to learn how the world works.

One consequence is that regret-bounded agents (Def. 5) are effectively limited to domains that are 'solvable', i.e. where we can feasibly learn a model of the underlying dynamics and use it to plan over long horizons. In domains where this is infeasible, there can be no guarantee the agent will generalize (satisfy a non-trivial regret bound $\delta < 1$) for long horizon tasks ($n \gg 1$). Therefore, some amount of online learning will be necessary, which is limited by the speed of interaction with the environment. Note that our results are derived for the simplest non-trivial environments (Assumption 1), and it is likely these constraints will be even stronger in more realistic environments which incorporate partially observed states or non-Markovian dynamics.

**Limitations.** The proof of Theorem 1 considers only fully observed environments. It is not clear what an agent operating in a partially observed environment would have to learn about latent variables in order to achieve the same level of behavioural flexibility. It is important to clarify that Theorem 1 proves the existence of a world model encoded in the agent's policy, not its specific use (e.g. for planning), nor can we make deeper epistemological claims about what the agent knows about its environment (Fagin et al., 2004).

## 5. Related work

**Inverse reinforcement learning** (IRL) (Ng et al., 2000) and inverse planning (Baker et al., 2007) involve determining an agents reward function (or goal) given the transition function and the optimal policy. Similarly, planning is the process of determining an optimal policy given the transition function and a goal (reward). Our result fills in the remaining direction, recovering the transition function given the agent's goal and their regret-bounded policy. In IRL the reward function can only be fully determined if we know the optimal policy across multiple environments (Amin & Singh, 2016), and likewise we find that to fully determine the environment transition function we must know the optimal policy for multiple goals. Figure 1 shows how our result relates to planning and IRL, where for each process takes as input two elements from {environment, goal, policy}, and determines the missing third element.

**Mechanistic Interpretability** (MI) aims to uncover implicit world models within model-free agents (Abdou et al., 2021; Li et al., 2022; Gurnee & Tegmark, 2023a; Karvonen, 2024; Hou et al., 2023; Bush et al., 2025). This typically involves learning a map from a policy network's activations to features representing states $S$ (e.g. the board states of a game (Li et al., 2022)). The state-space (ontology) $S$ is either assumed (as in supervised probing Alain & Bengio (2016)) or identified through unsupervised learning (as with SAEs Bricken et al. (2023)). The causal role of these features in the agent's decision making is established by intervening on their representations and observing the policy changes consistently, as if the world state had changed.

Our work also establishes an agent has learned a world model by the existence of a recovery map, but crucially this map is from the agent's policy rather than its activations. This is strictly weaker (as the policy is a function of the activations), and so Algorithm 1 can be used even when activations are inaccessible (e.g. private weights). This also allows us to tie the existence of a world model to agent capabilities (regret bounds as in Def. 5) rather than the specifics of the agent architecture, and Algorithm 1 applies

to all agents satisfying Def. 5 and environments satisfying Assumption 1. By comparison, probes or SAEs are fit to a given agent-environment system, and may require retraining if either changes (e.g. through distributional shifts or weight updates). Also Algorithm 1 is unsupervised, whereas MI methods are at least partially supervised, which can lead to ambiguity as to where the world model is encoded (in the agent, the probe, or jointly).

Another key difference is that we recover a predictive world model $\hat{P}_{ss'}(a)$ capturing environment dynamics, rather than simply a state space representation $\boldsymbol{S}$. However, our aim is to prove the agent has learned the actual environment dynamics up to an error bound, not to recover the subjective world model used by the agent to generate its actions. As discussed in the paragraph 'representation theorems' below, if we introduce additional consistency assumptions similar to those used MI[1], we can recover the agent's subjective world model. One drawback is that we may underestimate what an agent knows about its environment—e.g. agents could learn a world model but strongly violate Def. 5 (e.g. due to errors in planning), so Algorithm 1 isn't guaranteed to recover this world knowledge whereas methods like probing may succeed. However, Section 3.1 shows that at least in simple environments, our procedure can work well even when the regret bound is trivialised ($\delta = 1$).

**Causal world models.** (Richens & Everitt, 2024) provides a similar result to Theorem 1, showing that an agent capable of adapting to a sufficiently large set of distributional shifts must have learned a causal world model. Our work has a different focus: we study an agents ability to generalize to new goals (task generalization) rather than adapting to new environments (domain generalization). A surprising consequence of our result combined with Richens & Everitt (2024) is that domain generalization requires strictly more knowledge of the environment than task generalization. To see this, consider a setting where the state comprises two variables $S = X \times Y$ and $X \to Y$. We can construct an optimal goal-conditioned agent (Def. 4) given the transition function $P_{ss'}(a) = P(X_{t+1} = x', Y_{t+1} = y' \mid A_t = a, X_t = x, Y_t = y)$, as an optimal goal-conditioned policy can be determined by planning on this model. However, the causal relation between $X \to Y$ is non-identifiable from $P_{ss'}(a)$, i.e. almost all distributions $P_{ss'}(a)$ are compatible with both $X \to Y$ and $X \leftarrow Y$. Therefore, task generalization does not require knowledge of the causal relation between concurrent environment variables $X_t$ and $Y_t$

---

[1]Consistency in MI requires the agent adapts their behaviour following interventions on their world model. This amounts to assuming regret-bounded behavior under interventions, which is tantamount to assuming the agent has a causal world model to begin with (Richens & Everitt, 2024). Hence, using this kind of interventional consistency to establish that an agent has a world model risks circular reasoning.

whereas domain generalization does. That said, in the cMP setting the transition function does encode a degree of causal information, and we leave it to future work to determine precisely what causal knowledge is required for an agent satisfying Def. 5. This hints at an agential version of Pearl's causal hierarchy (Bareinboim et al., 2022), where different agent capabilities (like domain or task generalization) provably require different degrees of causal knowledge.

**LTL goal-conditioned agents.** LTL is the natural choice for expressing instructions, goals and safety constraints in reinforcement learning and planning (Camacho et al., 2019). Recently, there have been several implementations of goal-conditioned agents that generalize zero-shot to arbitrary LTL goals (Qiu et al., 2023; Jackermeier & Abate, 2025; Vaezipoor et al., 2021; Kuo et al., 2020). This maps precisely onto the setting we study, and future work could explore using Algorithm 1 or variants to recover world models from these agents, and use them to debug agent behaviour.

**Representation theorems** such as Savage (1972) and Halpern & Piermont (2024), establish that agents satisfying certain rationality axioms behave as if they are maximizing the expected value of a utility function with respect to a world model. For example, Savage (1972) can be used to 'fit' a world model to agent's behaviour, determining a unique utility function $U(s')$ and set of beliefs (a world model $\hat{P}_{ss'}(a)$) such that the policy that maximizes $\mathbb{E}_{\hat{P}}[U]$ is identical to the agents policy. However, this says nothing about what (if anything) the agent has learned about the true environment dynamics. For example, we may be able to assign a specific world model and utility function to a purely random policy $\pi(a \mid s) = 1/|\boldsymbol{A}|$, but this clearly does not imply that learning a world model is necessary to generate a random policy. Instead of attempting to recover an agent's subjective world model, we aim to recover the true underlying dynamics of the environment from the policy of the agent. In doing so, we show that learning such a policy implies learning these dynamics, and so the learnability of these dynamics bounds agent capabilities. Further, Theorem 2 establishes that an optimal myopic agent *does not* need to learn the transition probabilities $P_{ss'}(a)$, and representation theorems typically focus on the myopic regime.

We can recover something like the agent's subjective world model by changing Def. 5 to the assumption that the agent is $\delta$-optimal with respect to its own world model $\mathcal{M}$,

$$P_{\mathcal{M}}(\tau \models \psi \mid \pi, s_0) \geq \max_{\pi} P_{\mathcal{M}}(\tau \models \psi \mid \pi, s_0)(1 - \delta)$$
(3)

This amounts to assuming the agent has a world model, and that its behaviour is highly consistent with this world model (with consistency given by $\delta$), but stops short of assuming the agent is optimal with respect to its own beliefs. For example, $\delta > 0$ could represent a sub-optimal planner. For this altered Def. 5, Theorem 1 is unchanged and Algo-

rithm 1 returns the agents subjective world model $\mathcal{M}$ with bounded error. This may be appealing as a representation theorem as it has much weaker assumptions than Savage (1972), e.g. we only assume the agent follows a policy that is imperfectly consistent with its beliefs, whereas Savage (1972) requires the agent specifies a preference order over all actions (whereas a policy specifies only the most preferred action(s)), and makes strong rationality assumptions which are not satisfied by most current systems (Raman et al., 2024a).

**Good regulator theorem.** This influential theorem attempts to establish a similar result to ours, than any agent capable of controlling a system is in some sense a model of that system (Conant & Ross Ashby, 1970). However, as pointed out in (Wentworth, 2021), what the theorem actually shows is that, under several strong assumptions, an agent that minimizes the entropy of its environment must have a deterministic policy. This deterministic policy is then interpreted as a model of the environment, with the actions assigned to different states corresponding to a state representation. This is in spite of the fact that the policy (and hence the world model) could be a constant function, assigning the same action to every state. We do not consider an agent having a deterministic policy to be meaningful evidence that the agent has a model of its environment, and our theorem less ambiguously demonstrates that a world model capable of predicting the evolution of the environment has been learned by the agent.

**Theories of agency.** That agents have world models is a foundational assumption for several prominent theories in psychology and neuroscience; from constructivist theories of perception (Gregory, 1980) to active inference (Friston, 2010) and theories of consciousness (Safron, 2020). Like representation theorems, these theories aim to provide explanatory models of natural agents, rather than proving that agents necessarily conform to their assumptions. Our results offer a strong theoretical justification for these frameworks by demonstrating that goal-directed agents must acquire world models to achieve a degree of behavioural flexibility. Moreover, our findings remove the need to assume agents have world models a priori. Instead, we can assume a level of competency which implies their existence—arguably a more defensible position as competence can be measured.

## 6. Conclusion

The idea that the microstructure of an agent reflects the macrostructure of its environment is not new. It can be traced as far back as Democritus, who claimed that "man is a microcosm"—a miniature reflection of the cosmos (Allers, 1944)—and persists in contemporary scientific thinking— for example, Friston's assertion that "an agent does not have a model of it's world—it is a model" (Friston, 2013). While

this relation between agents and environments has long been hypothesised, we have sought to formalise and prove it. We have shown that any agent capable of generalizing to a sufficiently wide range of simple goal-directed tasks, must have learned an accurate model of it's environment. Essentially, all the information required to accurately simulate the environment is contained in the agent's policy. This implies that learning a world model is not only beneficial, but necessary for general agents. Consequently, efforts to create truly general AI cannot sidestep the challenge of world modeling, and instead should embrace it to unlock further capabilities and address critical issues in safety and interpretability.

Future work could extend our analysis to different classes of goals beyond Def. 3, and identify sets of simple 'universal' tasks that are sufficient to imply an agent has learned a world model. These tasks may then be useful for training general agents. Our results also point to new methods for inferring an agent's beliefs from their goals and behaviour without making strong rationality assumptions. Future work could build on Algorithm 1 to develop algorithms for recovering world models that are more scalable or apply to more general environments, and using these to improve agent safety and interpretability. On the more foundational level, Theorem 1 gives theoretical support to work in mechanistic interpretability looking to uncover implicit world models— for any agent capable of sufficiently general goal-directed behavior, the world model must be in there. Future work could use this necessity to derive new fundamental bounds on agent capabilities from the learnability of world models.

## Impact statement

This paper presents work whose goal is to advance the field of Machine Learning. There are many potential societal consequences of our work, none which we feel must be specifically highlighted here.

# A. Proof of Theorem 1

## A.1. Notation

In the following we denote random variables as capital letters $X$ and lower case letters $x$ denoting an event $X = x$ (equivalently, a value or state of $X$). We use bold letters to denote sets of variables $\boldsymbol{X} = \{X_1, X_2, \ldots, X_m\}$ and $\boldsymbol{x}$ denotes a set of events $\{x_1, x_2, \ldots, x_m\}$. We use square brackets to denote a proposition, e.g. $[X = x]$ returns True if $X = x$ and False otherwise. $I(p)$ denotes an indicator function, which returns 1 if the proposition $p$ is True and 0 if False.

## A.2. Environments

**Definition 1** (Controlled Markov process). *A controlled Markov process (cMP) is a Markov decision process (MDP) without a specified reward function or discount factor. It is defined by the tuple $(\boldsymbol{S}, \boldsymbol{A}, P_{ss'}(a))$ where $\boldsymbol{S}$ is the state space, $\boldsymbol{A}$ is the action space, and $P_{ss'}(a) = P(S = s' \mid A = a, S = s)$ is the transition function.*

First we assume the environment is described by a finite-dimensional, irreducible, controlled Markov process. For discussion of these standard assumptions see Puterman, 2014; Sutton, 2018.

**Assumption 1.** *We assume the environment is described by an irreducible, stationary, finite, controlled Markov process (Def. 1) with at least two actions.*

In the following we use $S = s_t$ and $A = a_t$ to denote the state of the environment and a the agent's choice of action at time $t$. The sequence of successive environment states and actions are referred to as a trajectory $\tau = (s_0, a_0, s_1, a_1, \ldots)$, with $\tau$ denoting an infinite length trajectory and we introduce an index $t_{i:j} = (s_i, a_i, \ldots, s_t, a_t)$ to denote a finite length trajectory between times $i$ and $j$. In some settings we use $h_{i:t} = (s_i, a_i, \ldots, s_t)$ to denote a finite length trajectory that should be interpreted as a history, i.e. a trajectory that has occurred, and which is truncated at $s_t$ (i.e. does not include $a_t$).

## A.3. Goals

Linear Temporal Logic (LTL) (Pnueli, 1977; Baier & Katoen, 2008) is a formalism widely used for expressing instructions, goals and safety constraints for agents (Littman et al., 2017; Li et al., 2017; Hasanbeig et al., 2019; Dzifcak et al., 2009; Ding et al., 2014). LTL extends classical propositional logic by introducing operators for reasoning about sequences of states over time, primary among them being,

- $\bigcirc$ (Next): The property holds in the next state,

- $\Diamond$ (Eventually): The property will hold at some point in the future,

- $\Box$ (Always): The property holds at every state from now on,

- $\mathcal{U}$ (Until): One property holds until another becomes true,

which can be combined with standard logical connectives (AND $\wedge$, OR $\vee$, NOT $\neg$ and material implication $\rightarrow$) to create complex goal specifications. The environment + agent system is described by the joint states $(s_t, a_t)$ where $s_t$ is the state of the environment and $a_t$ is the agent's action, at time $t$. Trajectories (paths) are a sequence of these states which we denote $\tau = (s_0, a_0, s_1, a_2, \ldots)$. An LTL expression $\varphi$ assigns a truth value to a given trajectory $\tau$, denoted $\tau \models \varphi$, which is true if $\tau$ satisfies $\varphi$ and false otherwise, with evaluation beginning at $t = 0$. For example, the trajectory of the environment-agent system $\tau = (s_0 = 0, a_0 = 0, s_1 = 1, a_0 = 0, \ldots)$ satisfies $\varphi = [s = 0] \wedge \bigcirc[s = 1]$ as the agent is in state $s = 0$ initially (at time $t = 0$) and in the next time step is in state $s = 1$.

Our desire is to define a minimal class of goals that describe the simplest and most intuitive goal-directed behaviours. To this end we focus on the most common definition of goals as being desirable states of the environment-agent system (Liu et al., 2022), which must be achieved within some time horizon.

**Definition 2** (Goals). *A goal $\varphi$ is an LTL expression of the form $\varphi = \mathcal{O}([(s, a) \in \boldsymbol{g}])$ where,*

- *$\boldsymbol{g}$ is a set of goal-states, a sub-set of the joint states of the environment-agent system $(s, a) \in \boldsymbol{S} \times \boldsymbol{A}$,*

- *$\mathcal{O}$ is a temporal operator specifying the time horizon for reaching $\boldsymbol{g}$. We restrict to $\mathcal{O} \in \{\bigcirc, \Diamond, \top\}$ where $\bigcirc$ = Next, $\Diamond$ = Eventually, $\top$ = Now.*

Rather than considering time horizons at the level of specific time indices $t$, which would require the agent to be capable of a high degree of environment control (e.g. 'reach state $S = s$ in precisely three time steps'), we focus on two simple time horizons; goals that are achieved immediately (now, $\top$), in the next time step (next, $\bigcirc$), or at any time in the future (eventually, $\Diamond$). Note that in LTL expressions the 'now' temporal operator is the identity, and we use $\top$ (True) to denote this. As $\top([X = x]) = [X = x]$ we suppress $\top$, e.g. $\varphi_i = [(s, a) \in \boldsymbol{g}_i]$, for ease of notation.

**Example:** consider the following goal for a cleaning robot: move eventually to the kitchen and in the next time step turn on the dish washer. This goal can be expressed as $\varphi = \Diamond([S = \text{in kitchen}] \wedge \bigcirc[A = \text{turn on dishwasher}])$. A trajectory $\tau$ satisfies this goal (denoted $\tau \models \varphi$) if $\exists\, t$ s.t. $S_t = \text{in kitchen}$ and $A_{t+1} = \text{turn on dishwasher}$

Going beyond the simplest, one-step goal-directed tasks requires an agent to achieve multiple sub-goals in a particular order. Our aim is to define a sequential goal $\psi$ in such a way that $\tau \models \psi$ is true if and only if each sub-goal state $\boldsymbol{g}_i$ is reached by the agent in the correct order. Expressing these sequential goals in LTL can be cumbersome, so for notational neatness we define a sequential goal formula $\psi = \langle \varphi_1, \ldots, \varphi_L \rangle$ which stands in for the more complex LTL expression, which is given by the the recursive formula in Def. 3. It will not be necessary to define sequential goals in general, as for our proofs we will focus on a simple class of sequential goals where the agent must reach a goal state either immediately or eventually.

**Definition 6** (Sequential goals). $\psi = \langle \varphi_1, \varphi_2, \ldots, \varphi_L \rangle$ *denotes the sequence of sub-goals (Def. 2)* $\varphi_1, \varphi_2, \ldots, \varphi_n$, *where* $\varphi_i = \mathcal{O}_i([(s, a) \in \boldsymbol{g}_i])$, $\mathcal{O}_i \in \{\Diamond, \top\}$ *and* $n = depth(\psi)$ *is the goal depth.* $\psi$ *can be expressed in linear temporal logic using the following recursive formula,*

$$\langle \varphi_1, \varphi_2, \ldots, \varphi_L \rangle = \begin{cases} [(s, a) \in \boldsymbol{g}_1] \wedge \langle \varphi_2, \ldots, \varphi_L \rangle, & \mathcal{O}_1 = \top \\ \bigcirc([(s, a) \in \boldsymbol{g}_1] \wedge \langle \varphi_2, \ldots, \varphi_L \rangle), & \mathcal{O}_1 = \bigcirc \\ [(s, a) \notin \boldsymbol{g}_1]\mathcal{U}([(s, a) \in \boldsymbol{g}_1] \wedge \langle \varphi_2, \ldots, \varphi_L \rangle), & \mathcal{O}_1 = \Diamond \end{cases} \tag{4}$$

*where* $\top = \text{True}$ *and* $\mathcal{O}_i = \top$ *denotes the Now (trivial) temporal operator, and for the singleton* $\langle \varphi \rangle = \varphi$.

By applying (4) recursively we can convert any sequential goal $\psi$ into an LTL expression. To understand (4) we can consider the simple case with two sub-goals $\psi = \langle \varphi_1, \varphi_2 \rangle$ and trajectory $\tau = (s_0, a_0, s_1, a_1, \ldots)$. If $\mathcal{O}_1 = \top$, then $\psi$ is satisfied if $(s_0, a_0) \in \boldsymbol{g}_1$ and the trajectory starting from the next time step $\tau_1 = (s_1, a_1, s_2, a_2, \ldots)$ satisfies $\varphi_2$. For $\mathcal{O}_1 = \Diamond$ the LTL expression we desire is $\langle \varphi_1, \varphi_2 \rangle = [(s, a) \notin \boldsymbol{g}_1]\mathcal{U}([(s, a) \in \boldsymbol{g}_1] \wedge \varphi_2)$. To see this, consider the case where $\mathcal{O}_2 = \top$, i.e. the agent's goal is to eventually reach $\boldsymbol{g}_1$, and then in the next time step to reach $\boldsymbol{g}_2$. If we attempt to express this goal as $\psi = \Diamond([(s, a) \in \boldsymbol{g}_1] \wedge \varphi_2)$, note that $\tau \models \psi$ if $\exists\, t$ s.t. $(s_t, a_t) \in \boldsymbol{g}_1$ and $(s_{t+1}, a_{t+1}) \in \boldsymbol{g}_2$. This includes trajectories where the agent reaches $\boldsymbol{g}_1$ and then fails to transition to $\boldsymbol{g}_2$ in the next time step, arbitrarily many times, so long as eventually the agent achieves the desired transition. Our aim is to express sequential goals where after satisfying a sub-goal $\varphi_i$ the agent switches to pursuing the sub-goal $\varphi_{i+1}$ in the next time step, and if the agent fails to satisfy this sub-goal then it fails to satisfy the overall sequential goal. The expression $[(s, a) \notin \boldsymbol{g}_1]\mathcal{U}([(s, a) \in \boldsymbol{g}_1] \wedge \varphi_2)$ enforces the condition $[(s, a) \notin \boldsymbol{g}_1]$ (the agent is not in goal-state $\boldsymbol{g}_1$) until they eventually reach $g_1$ at some $t$, and their trajectory commencing $t + 1$ satisfies $\varphi_2$, which captures the desired goal-switching behaviour.

**Example:** Consider the goal of transitioning eventually to $S = s$, then in the next time step transitioning to state $S = s'$ and then eventually returning to $S = s$. This is captured by the sequential goal $\psi = \langle \varphi_1, \varphi_2, \varphi_1 \rangle$ with sub-goals $\varphi_2 = \Diamond \boldsymbol{g}_1$ where $\boldsymbol{g}_1 = \{(a, s) \forall a \in \boldsymbol{A}\}$ and $\varphi_1 = \boldsymbol{g}_2$ where $\boldsymbol{g}_1 = \{(a, s') \forall a \in \boldsymbol{A}\}$. Applying (4) gives $\psi = [(s, a) \notin \boldsymbol{g}_1]\mathcal{U}([(s, a) \in \boldsymbol{g}_1] \wedge \bigcirc([(s, a) \in \boldsymbol{g}_2] \wedge \Diamond([(s, a) \in \boldsymbol{g}_1])))$, which is satisfied by any $\tau$ s.t. i) $\exists\, t$ s.t. $S_t = s$ and $S_{t'} \neq s \,\forall\, t' < t$, ii) $S_{t+1} = s'$ and ii) $\exists\, t' > t + 1$ s.t. $S_{t'} = s$.

Finally, we consider the case where there are multiple sequential goals the agent could satisfy, each corresponding to a different course of action that would be sufficient to achieve an overall goal. For example, a doctor's goal of providing primary care to a patient can be satisfied by several mutually exclusive pathways, such as providing a primary diagnosis and prescription, referring to a specialist for diagnosis, and so on. Each of these is its own task described by a sequence of sub-goals (e.g. attempting a primary diagnosis may involve question asking, performing an examination, etc), the outcome of which can inform the path the doctor takes (e.g. if an examination is inconclusive, they may refer to a specialist). Each of these pathways therefore corresponds to a different sequential goal, and satisfying any of these sequential goals satisfies the overall goal of providing care to the patient.

To formalise this we consider goals that are disjunctions over multiple sequential goals. Let $\boldsymbol{\Psi}$ denote the set of all *composite* goals, which includes all disjunctions over all sequential goals (Def. 6), i.e. $\psi, \psi' \in \boldsymbol{\Psi} \implies \psi \vee \psi' \in \boldsymbol{\Psi}$. For a conjunction

over goals $\psi'' = \psi \vee \psi'$, then agent satisfies $\psi''$ if its policy generates a trajectory $\tau = (s_0, a_1, s_1, a_2, \ldots)$ that satisfies $\psi$ or $\psi'$.

**Definition 3** (Composite goals). *A sequential goal $\psi$ is an ordered sequence of sub-goals (Def. 2) $\psi = \langle \varphi_1, \ldots \varphi_n \rangle$, where the agent must achieve sub-goal $\varphi_i$ before $\varphi_{i+1}$. The depth of a sequential goal is the number of sub-goals $depth(\psi) = n$. A composite goal is a disjunction of one or more sequential goals $\psi = \bigvee_{i=1}^{m} \psi_i$, i.e. the agent must achieve any sub-goal $\psi_i$ to achieve $\psi$. The depth of a composite goal is the max depth of its sub-goals $depth(\psi) = \max_{\psi_i} depth(\psi_i)$. $\boldsymbol{\Psi}_n$ is the set of all composite goals $\psi$ with $depth(\psi) \leq n$.*

**Example:** Consider the simple navigation task where a robot cleaner is required to clean the kitchen and the living room in any order, and then return to its charging station. There are two pathways that satisfy this; 1) clean the kitchen (eventually), then clean the living room (eventually), then return to the charging point (eventually), 2) the same as 1) but with the kitchen and living room swapped. Formally, the robot satisfies the overall goal if it generates a trajectory $\tau$ that satisfies the LTL expression $\psi = \psi_1 \vee \psi_2$ where $\psi_1 = \langle \varphi_1, \varphi_2, \varphi_3 \rangle$, $\psi_1 = \langle \varphi_2, \varphi_1, \varphi_3 \rangle$, $\varphi_1 = \Diamond([(s,a) \in \boldsymbol{g}_1 = \{(s_1, a_1)\}])$, $\varphi_2 = \Diamond([(s,a) \in \boldsymbol{g}_2 = \{(s_2, a_1)\}])$, $\varphi_3 = \Diamond([s \in \boldsymbol{g}_3 = \{s_3\}])$, $s_1 = $ in kitchen, $s_2 = $ in livingroom, $s_3 = $ at charging station and $a_1 = $ clean.

## A.4. Agents

We assume the environment is described by a cMDP (Def. 1). Due to the generally non-Markovian nature of sequential goals, we consider the most general definition of agents as maps from histories and goals to actions. A goal conditioned agent is a policy $\pi(a_t \mid h_t; \psi)$, where $\psi$ is a (composite) goal. For simplicity we restrict our attention to agents that follow deterministic policies. In general the environment may evolve non-deterministically, so the objective is to maximise the probability that $\tau \models \vee \psi$, which is determined by summing over the probabilities of all trajectories that could result from $\pi$ and that satisfy $\psi$ (Qiu et al., 2024).

**Definition 4** (optimal goal-conditioned agent). *For a given set of goals $\boldsymbol{\Psi}$ (Def. 3) an optimal agent is a goal-conditioned policy $\pi^*(a_t \mid h_t; \psi)$ where $\pi^*$ is deterministic and satisfies,*

$$\pi^* = \arg\max_{\pi} P(\tau \models \psi \mid \pi, s_0) \tag{1}$$

$\forall \, s_0$ *s.t.* $P(s_0) > 0$*, where $s_0$ is the initial state of the environment at $t = 0$, and $\forall \, \psi \in \boldsymbol{\Psi}$.*

$P(\tau \models \psi \mid \pi, s_0)$ is the probability that the trajectory $\tau$ generated by the agent under policy $\pi$ satisfies the composite goal $\psi$ (LTL expression is given by Def. 3),

$$P(\tau \models \psi \mid \pi, s_0) = \sum_{\tau} P(\tau \mid \pi, s_0) I([\tau \models \psi]) \tag{5}$$

In other words, an optimal goal-conditioned agent can achieve any composite goal $\psi \in \boldsymbol{\Psi}$ with the maximum probability of success attainable for every initial state $S_0 = s_0$ that the agent could start in.

It is of course unreasonable to assume that any realistic agent is capable of optimally satisfying any given composite goal $\psi$ in its environment, and so we consider two relaxations of Def. 4; sub-optimal agents, and restricted the complexity of the goals the agent is capable of achieving.

Firstly, the most intuitive way to bound the agent's optimality carries over from regret bounds in reinforcement learning (Sutton, 2018), but instead of providing a lower bound on the cumulative discounted reward compared to the optimal agent, we can lower bound on the probability that the agent achieves a given goal compared to the optimal agent. Secondly, achieving goals that involve a larger number of sub-goals (a higher goal depth $n$, Def. 3) is more difficult than achieving short-term or myopic goals, and intuitively requires more knowledge of the environment. For example, if we restricted to one-step goals ($\mathcal{O}_i = \top$ and $n = 1$), simply knowing $\arg\max_a P_{ss'}(a)$ would be sufficient to identify an optimal policy, thus a full world model capable of simulating the environment is clearly not required. On the other hand, if an agent uses a world model to plan, effectively planning for longer sequences of sub-goals requires an increasingly accurate model, as errors compound over time. Hence, in deriving our results it is natural to consider agents with some bounded maximum goal depth $n$, such that there is no guarantee that agent can satisfy the regret bound for sequential goals with depth greater than $n$.

To this end, we propose the following definition of a bounded goal-conditioned agent defined by two parameters; $\delta$ (the lower bound on the probability of achieving a goal compared to an optimal agent), and $n$ (the maximum goal depth for which the $\delta$ bound applies).

**Definition 5** (bounded goal-conditioned agent). *A bounded goal-conditioned agent is a goal-conditioned policy $\pi(a_t \mid h_t; \psi)$ satisfying,*

$$P(\tau \models \psi \mid \pi, s_0) \geq \max_{\pi} P(\tau \models \psi \mid \pi, s_0)(1 - \delta) \tag{2}$$

$\forall \, \psi \in \mathbf{\Psi}_n$ *where $n$ is the maximum goal depth and $s_0$ is the initial state of the environment at $t = 0$.*

### A.5. Overview of proof of Theorem 1

At a high level, the proof of Theorem 1 can be understood as deriving an algorithm that estimates $P_{ss'}(a)$ by querying the bounded agent's policy $\pi(a_t \mid h_t, \psi)$ with different composite goals and observing how the agent's action choice changes. We consider composite goals where the agent is required to navigate (eventually) to a specific state $S = s$ and take an action $A = a$, transitioning to an outcome state, and then returns eventually to $S = s$ (Figure 4). We compare two goals, the first $\psi_1(r, n)$ which is satisfied if the outcome state is $S = s$ at most $r$ times, out of a total of $n$ trials (taking action $A = a$ in $S = s$), and the second $\psi_2(r, n)$ where the outcome is $S = s$ at least $r + 1$ times. An optimal agent can achieve the first

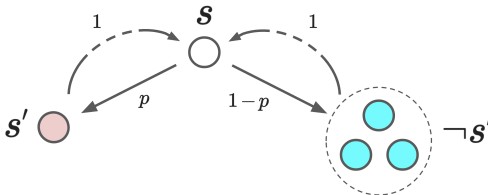

Figure 4: Figure illustrates the composite goal in the proof of Theorem 1.

goal with a probability given by the cumulative binomial distribution $P_b(X \leq r)$ where $X$ is the total number of 'successful' transitions $(a, s) \to s'$, which occur with probability $P_{ss'}(a)$, and likewise the second goal can be achieved with probability $P_b(X > r)$. Hence, as we increase $r$ from 0 to $n$, an optimal agent will switch from pursuing the second goal to pursuing the first goal when $r$ reaches a value that exceeds the median number of successes, and we show it is possible to identify this 'goal switching' in the agent's policy $\pi(a_t \mid h_t, \psi_1(r, n) \vee \psi_2(r, n))$. The median is given by $\lfloor P_{ss'}(a)(n + 1) \rfloor$ and so we can bound $P_{ss'}(a)$ with an error that scales as $1/n$. For $\delta > 0$, the goal-switching behaviour of the agent cannot precisely determine the median, but bounds it within a region, and this allows us to approximate $P_{ss'}(a)$ with an error that depends on $\delta$.

### A.6. Proof of Theorem 1

We now prove our main results.

**Lemma 1.** *For a finite dimensional, stationary and irreducible cMPD (Assumption 1) there exists a deterministic Markovian policy $\pi_{s'}(a \mid h = (s_0, a_0, \ldots, s_T)) = \pi_{s'}(a \mid s_T)$ that eventually reaches a given state $S = s'$ from any other state $S = s$ with probability 1.*

*Proof.* Irreducibility states that for any $s' \neq s$ there exists a finite sequence of actions that reaches any $S = s'$ from any $S = s$ with non-zero probability. Therefore, for any $S = s'$ we can construct a tree of states by; i) starting with the root $s'$ and defining the set $\mathbf{Z} = \mathbf{S} \setminus \{s'\}$, ii) for each $s'' \in \mathbf{Z}$, if $\exists \, A = a''$ s.t. $P_{s''s'}(a'') > 0$ then $s''$ is a parent of $s'$ in the tree and we remove $s''$ from $\mathbf{Z}$, iii) repeat for all parents of $s'$ and so on, until $\mathbf{Z} = \emptyset$. As the cMDP is finite dimensional and irreducible, the resulting tree traverses the state space and is of finite depth, and by construction every state in the tree $s_i$ has a single child $s_j$ and the tree contains no loops. For each $s_i$ we can associate an action $a(s_i)$ given by $a(s_i) = \arg\max_a P_{s_i s_j}(a)$. Consider the Markovian policy $\pi(A = a \mid h = (s_0, a_0, \ldots, s_T) = [a = a(s_T)]$, which attempts to move from the most recent state $s_T$ to $s'$ by traversing the tree. For every state, there is a non-zero probability that $\pi$ succeeds in traversing the tree to the root $S = s'$. If the agent fails a given transition $S_t = s_i \to S_{t+1} = s_j$, the process of traversing the tree begins again from $S_{t+1}$, and as the policy and environment are Markovian, each attempt to

traverse to $S = s'$ is independent. Hence, $\pi$ attempts to reach $S = s'$ with an unbounded number of independent trials, each with non-zero probability of success, and hence eventually reaches $S = s'$ with probability 1.

For $s' = s$, the problem is identical except that the deterministic policy can take any action in $S = s$. Let $S = s_1$ be the state that this transitions to. If $s_1 = s$ then the policy has reached $S = s$. If $s_1 \neq s$ we follow the deterministic Markovian policy derived in the previous section which eventually reaches $S = s$ with probability 1. $\qquad\square$

The following lemma allows us to simplify our analysis by letting us consider optimal policies in environments with extended action spaces, where determining optimal policies is easier.

**Lemma 2.** *Consider extending the action space of the environment cMDP with a single action $A = \bar{a}$, $\boldsymbol{A}' = \boldsymbol{A} \cup \{\bar{a}\}$ where the extended transition function $P'$ has $P'_{ss'}(a) = P_{ss'}(a) \; \forall \, a \neq \bar{a}$, and $P'_{ss'}(\bar{a})$ is any valid conditional probability distribution. For any given composite goal $\psi$ the optimal policy for the extended action space $\boldsymbol{A}'$ achieves $\psi$ with a probability greater or equal to that for the optimal policy in the unextended action space $\boldsymbol{A}$,*

$$\max_{\pi} P'(\tau \models \psi \mid \pi, s_0) \geq \max_{\pi} P(\tau \models \psi \mid \pi, s_0)$$

*Proof.* Let $P'_{ss'}(a)$ denote the new transition function in the extended environment. As $P'_{ss'}(a) = P_{ss'}(a) \; \forall \, a \neq \bar{a}$, the probability of any $\pi$ that does not take action $A = \bar{a}$ satisfying any given composite goal $\psi$ is the same in the extended and unextended environments. As the optimal policy $\pi^*_{\boldsymbol{A}}$ over $\boldsymbol{A}$ does not take action $A = \bar{a}$ (as $\bar{a} \notin \boldsymbol{A}$), then

$$P'(\tau \models \psi \mid \pi^*_{\boldsymbol{A}}, s_0) = P(\tau \models \psi \mid \pi^*_{\boldsymbol{A}}, s_0) \tag{6}$$

Therefore there exists a policy for the extended action space (namely $\pi^*_{\boldsymbol{A}}$) that achieves $\psi$ with the same probability as the optimal policy for the unextended action space. Therefore,

$$\max_{\pi} P'(\tau \models \psi \mid \pi, s_0) \geq \max_{\pi} P(\tau \models \psi \mid \pi, s_0) \tag{7}$$

$\qquad\square$

We now derive Lemmas that let us factor and simplify sequential goals.

**Lemma 3.** *For $\psi = \langle \varphi_1, \ldots, \varphi_L \rangle$ and a cMP obeying Assumption 1, if $\varphi_1 = \Diamond([S = s_g, A = a_g])$ and $\pi(a_t \mid h_t) = \pi(a_t \mid s_t)$ is a stationary Markovian policy that eventually reaches $S = s_g$ and takes $A = a_g$ from $S_0 = s_0$ with probability 1, then,*

$$P(\tau \models \langle \varphi_1, \varphi_2, \ldots, \varphi_L \rangle \mid \pi, s_0) = P(\tau \models \langle \varphi_2, \ldots, \varphi_L \rangle \mid \pi, s_g)$$

*Proof.* Using Def. 3 we can simplify $\langle \varphi_1, \varphi_2, \ldots, \varphi_L \rangle = [S \neq s_g] \mathcal{U}([S = s_g] \wedge \langle \varphi_2, \ldots, \varphi_L \rangle)$. If $s_0 = s_g$ then $\varphi_1$ is satisfied at $t = 0$ and $\varphi_1$ is trivialised, i.e. $P(\tau \models \langle \varphi_1, \varphi_2, \ldots, \varphi_L \rangle \mid \pi, s_g) = P(\tau \models \langle \varphi_2, \ldots, \varphi_L \rangle \mid \pi, s_g)$. Therefore we need only consider the case where $s_0 \neq s_g$.

As $\pi$ reaches $S = s_g$ from $S_0 = s_0$ with probability 1, every trajectory generated by $\pi$ eventually reaches $S = s_g$ by assumption, and at some $T > 0$ as $s_0 \neq s_g$. For a given $\tau$ let $T$ be the time step that $\tau$ first reaches $s_g$. Because $\pi$ is deterministic and Markovian, and the environment is Markovian, then for $s_0 \neq s_g$ we can express,

$$P(\tau \mid s_0, \pi) = \prod_{i=1}^{T} P(s_i \mid s_{i-1}, \pi'(s_{i-1})) P(\tau_{T+1} \mid S_T = s_g, \pi'(S_t = s_g)) \tag{8}$$

where $\pi'$ is the policy for $t > 0$, and as $\pi'$ is stationary we have that $P(\tau_{T+1} \mid S_T = s_g, \pi'(S_t = s_g)) = P(\tau_{T+1} \mid$

$s_g, \pi'(s_g))$. Let $h_T$ be the trajectory up to $S_T = s_g$. Using the LTL expression for $\psi$ from Def. 6 gives,

$$P(\tau \models \langle \varphi_1, \varphi_2, \ldots, \varphi_L \rangle \mid \pi, s_0) = \sum_{h_T} P(h_T \mid s_0, \pi) \sum_{\tau_{T+1}} P(\tau_{T+1} \mid h_T, \pi(h_T)) I([\tau \models [S \neq s_g]\mathcal{U}([S = s_g] \wedge \langle \varphi_2, \ldots, \varphi_L \rangle)])$$

$$(9)$$

$$(10)$$

$$= \sum_{h_T} P(h_T \mid s_0, \pi) I([S \neq s_g]\mathcal{U}[S = s_g]) \sum_{\tau_{T+1}} P(\tau_{T+1} \mid h_T, \pi(h_T)) I([\tau_T \models \langle \varphi_2, \ldots, \varphi_L \rangle])$$

$$(11)$$

$$= \sum_{h_T} P(h_T \mid s_0, \pi) \sum_{\tau_{T+1}} P(\tau_{T+1} \mid s_g, \pi(s_g)) I([\tau_T \models \langle \varphi_2, \ldots, \varphi_L \rangle]) \tag{12}$$

$$= \sum_{h_T} P(h_T \mid s_0, \pi) P(\tau_{T+1} \models \langle \varphi_2, \ldots, \varphi_L \rangle \mid S_T = s_g, A_t = \pi(s_g)) \tag{13}$$

$$= P(\tau \models \langle \varphi_2, \ldots, \varphi_L \rangle \mid s_g, \pi)) \tag{14}$$

where in the last line we have used $\sum_{h_T} P(h_T \mid \pi, s_0) = 1$ by assumption (as $\pi$ reaches $S = s_g$ from $S_0 = s_0$ with probability 1). $\qquad\square$

**Lemma 4.** *For $\psi = \langle \varphi_1, \varphi_2, \varphi_3, \ldots, \varphi_L \rangle$ and a cMP obeying Assumption 1, if $\varphi_1 = \bigcirc([s \in \mathbf{g}_1])$ and $\varphi_2 = \Diamond([S = s_g, A = a_g])$ and $\pi$ is a deterministic, Markovian policy then*

$$P(\tau \models \psi \mid s_0, \pi) = P(S_1 \in \mathbf{g}_1 \mid s_0, \pi) P(\tau \models \langle \varphi_3, \ldots, \varphi_L \rangle \mid s_g, \pi)$$

*Proof.* Using Def. 6 and $\tau_k = (s_k, a_k, \ldots)$ we get,

$$P(\tau \models \psi \mid s_0, \pi) = \sum_{\tau_1} P(\tau_1 \mid s_0, a_0 = \pi(s_0)) I([\bigcirc([s \in \mathbf{g}_1] \wedge \langle \varphi_2, \ldots, \varphi_L \rangle)]) \tag{15}$$

$$= \sum_{\tau_1} P(\tau_1 \mid s_0, a_0 = \pi(s_0)) I([s_1 \in \mathbf{g}_1] \wedge [\tau_1 \models \langle \varphi_2, \ldots, \varphi_L \rangle]) \tag{16}$$

$$= \sum_{s_1} P(s_1 \mid s_0, a_0 = \pi(s_0)) I([s_1 \in \mathbf{g}_1]) \sum_{\tau_2} P(\tau_2 \mid s_0, a_0 = \pi(s_0), s_1, a_1 = \pi(s_1)) I([\tau_1 \models \langle \varphi_2, \ldots, \varphi_L \rangle])$$

$$(17)$$

$$= \sum_{s_1} P(s_1 \mid s_0, a_0 = \pi(s_0)) I([s_1 \in \mathbf{g}_1]) \sum_{\tau_2} P(\tau_2 \mid s_1, a_1 = \pi(s_1)) I([\tau_1 \models \langle \varphi_2, \ldots, \varphi_L \rangle]) \tag{18}$$

$$= \sum_{s_1} P(s_1 \mid s_0, a_0 = \pi(s_0)) I([s_1 \in \mathbf{g}_1]) P(\tau_2 \models \langle \varphi_2, \ldots, \varphi_L \rangle \mid s_1, a_1 = \pi(s_1)) \tag{19}$$

$$= \sum_{s_1} P(s_1 \mid s_0, a_0 = \pi(s_0)) I([s_1 \in \mathbf{g}_1]) P(\tau \models \langle \varphi_3, \ldots, \varphi_L \rangle \mid s_g, \pi) \tag{20}$$

$$= P(S_1 \in \mathbf{g}_1 \mid s_0, \pi) P(\tau \models \langle \varphi_3, \ldots, \varphi_L \rangle \mid s_g, \pi) \tag{21}$$

where in line (20) we apply Lemma 3

$$\square$$

**Lemma 5.** *For $\psi = \langle \varphi_1, \varphi_2, \ldots, \varphi_L \rangle$ and a cMP obeying Assumption 1, if $\varphi_1 = \top([A = a])$ and $\pi$ is a deterministic policy s.t. $\pi(s_0) = a$ then $P(\tau \models \psi \mid s_0, \pi) = P(\tau \models \langle \varphi_2, \ldots, \varphi_L \rangle \mid s_0, \pi)$*

*Proof.* This follows simply from Def. 6 and that the policy is deterministic,

$$P(\tau \models \psi \mid s_0, \pi) = \sum_{\tau_1} P(\tau_1 \mid s_0, a_0 = \pi(s_0)) I([\tau \models [A_0 = a] \wedge \langle \varphi_2, \ldots, \varphi_N \rangle) \tag{22}$$

$$= \sum_{\tau_1} P(\tau_1 \mid s_0, a_0 = a = \pi(s_0)) I([\pi(s_0) = a]) I([\tau \models \langle \varphi_2, \ldots, \varphi_N \rangle) \tag{23}$$

$$= P(\tau \models \langle \varphi_2, \ldots, \varphi_L \rangle \mid s_0, \pi) \tag{24}$$

where $\tau_k = (s_k, a_k, \dots)$. $\qquad\qquad\qquad\qquad\qquad\qquad\qquad\qquad\qquad\qquad\qquad\qquad\qquad\qquad$ $\square$

We now derive a family of composite goals for which the optimal policy satisfies the goal with a probability given by the cumulative binomial distribution for which the probability parameters is a specific transition probability.

**Lemma 6.** *Let $\psi(r, n)$ be the composite goal which is the disjunction over all sequential goals of the form*

$$\psi = \langle \varphi_1, \underbrace{\varphi_2, \varphi_3, \varphi_2, \varphi_3, \dots \varphi_2, \varphi_3'}_{n \text{ times}} \rangle$$

*where the agent*

i) *takes action $A = b$, $\varphi_1 = [A = b]$, and then transitions eventually to $S = s$ and takes action $A = a$, $\varphi_2 = \Diamond([S = s, A = a])$,*

ii) *transitions next to a goal state which is either $S = s'$, $\varphi_3 = \bigcirc[S = s']$, or $S \neq s'$, $\varphi_3' = \bigcirc[S \neq s']$,*

iii) *returns eventually to $S = s$ and takes action $A = a$, $\varphi_2$, and repeats the cycle ii)-iii) a total of $n$ times, with the transition $\varphi_3 = [S' = s]$ occurring $r$ times and the transition $\varphi_3' = [S \neq s']$ occurring $n - r$ times.*

*For $s \neq s'$, the optimal policy achieves this goal with probability,*

$$\max_\pi P(\tau \models \psi(r, n) \mid \pi, s_0) = \frac{n!}{(n-r)!r!} P_{ss'}(a)^r (1 - P_{ss'}(a))^{n-r} \tag{25}$$

*Proof.* Let $\psi = \bigvee_i \psi_i$. Each $\psi_i$ involves a specific ordering of the $r$ sub-goals $\varphi_3 = [S = s]$ and the $n - r$ sub-goals $\varphi_3' = [S \neq s']$, hence they are mutually exclusive, i.e. $\nexists \tau$ such that $[\tau \models \psi_i] \wedge [\tau \models \psi_j]$ for any $\psi_i, \psi_j$ in the disjunction s.t. $\psi_i \neq \psi_j$, and hence,

$$P(\tau \models \psi(r, n) \mid \pi, s_0) = \sum_i P(\tau \models \psi_i \mid \pi, s_0) \tag{26}$$

First we evaluate $\max_\pi P(\tau \models \psi(r, n) \mid \pi, s_0)$ in the environment with the extended action space (Lemma 2) with $\boldsymbol{A}' = \boldsymbol{A} \cup \{\bar{a}\}$, $P'_{s''s}(\bar{a}) = 1 \, \forall \, s'' \in \boldsymbol{S}$ and $P'_{ss'}(a \neq \bar{a}) = P_{ss'}(a)$. I.e. we extend with the action $A = \bar{a}$ which returns the agent to $S = s$ from any state with probability 1. Note that until the agent has returned to $S = s$ a total of $n$ times, in order to satisfy any sequential goal $\psi_i$ comprising the composite goal $\psi(r, n)$ the agent must take action $A = b$ at $t = 0$ and $A = a$ when it is in $S = s$. The only freedom left to the agent is how it returns to $S = s$ (to satisfy $\varphi_2$) from whatever state it transitions to after taking $A = a$ in $S = s$, and for the extended action space it can achieve this immediately with probability 1 simply by taking action $A = \bar{a}$. Therefore, the following policy is optimal for satisfying $\psi(n, r)$ with the extended action space,

$$\bar{\pi}^*(A = a' \mid h = (s_0, a_0, \dots, s_t)) = \begin{cases} I([a' = b]), & t = 0 \\ I([a' = a] \wedge [s_t = s]) + I([a' = \bar{a}] \wedge [s_t \neq s]), & t > 0 \end{cases} \tag{27}$$

i.e. the agent first takes action $A = b$ (required to satisfy $\varphi_1$), and from then on it takes action $A = a$ in $S = s$ (required for $\varphi_3$ and $\varphi_3'$) and $A = \bar{a}$ otherwise, which returns the agent immediately to $S = s$. Applying Lemma 5 and Lemma 3 allows us to eliminate the first $\varphi_1$ and $\varphi_2$, giving

$$P'(\tau \models \psi_i \mid \bar{\pi}^*, s_0) = P'(\tau \models \langle \underbrace{\varphi_2, \varphi_3, \dots, \varphi_2, \varphi_3'}_{r \times \varphi_2, \varphi_3 \text{ and } (n-r) \times \varphi_2, \varphi_3'} \rangle \mid \bar{\pi}^{*\prime}, s) \tag{28}$$

where $\bar{\pi}^{*\prime}$ is $\bar{\pi}^*$ for $t > 0$, which we denote $\bar{\pi}^*$ from now on for ease of notation, and can treat $\bar{\pi}*$ as a stationary policy. Repeatedly applying Lemma 4 to (28) gives,

$$P'(\tau \models \psi_i \mid \bar{\pi}^*, s_0) = P_{ss'}(a)^r (1 - P_{ss'}(a))^{n-r} \tag{29}$$

Applying (26) and noting $P'(\tau \models \psi_i \mid \bar{\pi}^*, s_0) = P'(\tau \models \psi_j \mid \bar{\pi}^*, s_0)$ for all $i, j$ for $\psi(r, n) = \bigvee_i \psi_i$, and the total number of sequential goals comprising $\psi(r, n)$ is given by the number of combinations of size $r$ from $n$ objects ($n$ transitions, $r$ of which are $s \to s'$), and so we recover,

$$\max_\pi P'(\tau \models \psi(r, n) \mid \pi, s_0) = \frac{n!}{(n - r)! r!} P_{ss'}(a)^r (1 - P_{ss'}(a))^{n-r} \tag{30}$$

Finally, we construct a policy $\tilde{\pi}$ in the original (unextended) environment, and show that this saturating the upper bound in (30) and so by Lemma 2 is optimal, therefore Equation (25) holds. By Lemma 1 there exists a deterministic Markovian policy $\pi_{s'}(a \mid s)$ that transitions eventually to $S = s$ from any state with probability 1. Let,

$$\tilde{\pi}(a_t \mid s_t) = \begin{cases} I([a' = b]), & t = 0 \\ \pi_{s'}(a \mid s_t), & t > 0 \text{ and } s_t \neq s \\ I([a' = a]), & t > 0 \text{ and } s_t = s \end{cases} \tag{31}$$

Note $\tilde{\pi}$ is identical to $\bar{\pi}^*$ except that instead of taking action $A = \bar{a}$ in $S = s$ (as this action does not exist) the agent follows $\pi_{s'}$. As $\pi_{s'}$ is deterministic, stationary and Markovian, and eventually reaches $S = s$ with probability 1 from any $S = s'$, so we can apply Lemma 5 and Lemma 4 as before giving,

$$P(\tau \models \psi_i \mid \tilde{\pi}, s_0) = P_{ss'}(a)^r (1 - P_{ss'}(a))^{n-r} \tag{32}$$

which saturates the upper bound implied by (30) and Lemma 2, hence $\tilde{\pi}$ is optimal, and using $nCr = n!/((n - r)! r!)$ we recover (25). $\qquad\square$

We are now in a position to prove our main theorem.

**Theorem 1.** *Let $P_{ss'}(a) = P(S_{t+1} = s' \mid A_t = a, S_t = s)$ be the transition probabilities of an environment satisfying Assumption 1. Let $\pi$ be a goal-conditioned agent (Def. 5) with a maximum failure rate $\delta$ for all goals $\psi \in \Psi_n$ where $\Psi_n$ is the set of all composite goals with maximum goal depth $n > 1$. $\pi$ fully determines a model for the environment transition probabilities $\hat{P}_{ss'}(a)$ with errors satisfying*

$$\left| \hat{P}_{ss'}(a) - P_{ss'}(a) \right| \leq \sqrt{\frac{2 P_{ss'}(a)(1 - P_{ss'}(a))}{(n - 1)(1 - \delta)}}$$

*for any $n, \delta$, and for $\delta \ll 1$, $n \gg 1$ the error scales as,*

$$\left| \hat{P}_{ss'}(a) - P_{ss'}(a) \right| \sim \mathcal{O}\left(\delta/\sqrt{n}\right) + \mathcal{O}(1/n)$$

*Proof in Appendix A.6.*

*Proof.* Let $\psi_{a'}(k, n)$ denote composite goal as in Lemma 6 which is a disjunction over all sequential goals of the form

$$\psi = \langle \varphi_0, \underbrace{\varphi_1, \varphi_2, \ldots \varphi_1, \varphi_2'}_{n \text{ times}} \rangle \tag{33}$$

where the agent

i) takes action $A = a$ ($\varphi_0 = [A = a]$), and then transitions eventually to $S = s$ and takes action $A = a$ ($\varphi_1 = \Diamond([S = s, A = a])$),

ii) transitions next to a goal state which is either $S = s'$ ($\varphi_2 = \bigcirc[S = s']$) or $S \neq s'$ ($\varphi_2' = \bigcirc[S \neq s']$),

iii) returns eventually to $S = s$ and takes action $A = a$ ($\varphi_1$), and repeats the cycle ii)-iii) a total of $n$ times, with the transition $\varphi_2 = \bigcirc[S' = s]$ occurring $r$ times and the transition $\varphi_2' = \bigcirc[S \neq s']$ occurring $n - r$ times, for all $r \leq k$.

I.e. the agent's goal is to first take action $A = a$ and then to achieve the transition $(a, s) \to s'$ at most $k$ times out of $n$ attempts. Note that $n$ attempts corresponds to a goal depth of $2n + 1$.

Consider the sequential goals $\psi_b(k, n)$ that is identical to $\psi_a(k, n)$ except that the first sub-goal i) takes action $A = b$ instead of $A = a$ at time $t = 0$, and in iii) we have $r > k$ instead of $r \leq k$. I.e. the agents goal is to first take action $A = b$ and then to achieve the transition $(a, s) \to s'$ more than $k$ times out of $n$ attempts.

Consider the composite goal $\psi_{a,b}(k, n) = \psi_a(k, n) \vee \psi_b(k, n)$ for any pair of action $a, b$ such that $a \neq b$ (we assume there are at least two distinct action in Assumption 1).

Note that $\psi_a(k, n)$ and $\psi_b(k, n)$ are mutually exclusive, $\tau \models \psi_a(k, n) \implies \tau \not\models \psi_b(k, n)$ and vice versa, hence,

$$P(\tau \models \psi_{a,b}(k, n) \mid \pi, s_0) = P(\tau \models \psi_a(k, n) \mid \pi, s_0) + P(\tau \models \psi_b(k, n) \mid \pi, s_0) \tag{34}$$

and for any $\pi$ only one of the terms on the right hand side is non-zero. Hence we can evaluate $\max_\pi P(\tau \models \psi_a(k, n) \mid \pi, s_0)$ and $\max_\pi P(\tau \models \psi_b(k, n) \mid \pi, s_0)$ separately.

Consider a bounded goal-conditioned agent (Def. 5). As the policy is deterministic by assumption, the agent is can only choose one of two sub-goals to attempt to satisfy, $\psi_a(k, n)$ or $\psi_b(k, n)$, depending on its first action choice $A_0$. If $\pi(a_0 \mid s_0) = I([a_0 = a])$ then the agent is pursuing $\psi_a(k, n)$. For $\psi_a(k, n) = \bigvee_i \psi_i$ all $\psi_i, \psi_j$ are mutually exclusive for $\psi_i \neq \psi_j, \tau \models \psi_i \implies \tau \not\models \psi_j$ and vice versa, and hence,

$$P(\tau \models \psi_a(k, n) \mid \pi, s_0) = \sum_i P(\tau \models \psi_i \mid \pi, s_0) \tag{35}$$

and by Lemma 6 the maximum probability that this goal can be satisfied is given by,

$$\max_\pi P(\tau \models \psi_a(k, n) \mid \pi, s_0) = \sum_{r=0}^{k} P_n(X = r) = P_n(X \leq k) \tag{36}$$

where

$$P_n(X = r) := \frac{n!}{(n - r)!r!} P_{ss'}(a)^r (1 - P_{ss'}(a))^{n-r} \tag{37}$$

is the binomial probability mass function and $P_n(X \leq k)$ is the cumulative distribution function.

Likewise if $\pi(a_0 \mid s_0) = I([a_0 = b])$ the agent is pursuing $\boldsymbol{\psi}_b(k, n)$, which can be achieved with a maximum probability,

$$\max_\pi P(\tau \models \psi_b(k, n) \mid \pi, s_0) = \sum_{r=k+1}^{n} P_n(X = k) = P_n(X > k) \tag{38}$$

Finally if $\pi(a_0 \mid s_0) = I([a_0 = a'])$ where $a' \notin \{a, b\}$ then the agent satisfies $\psi_{a,b}(k, n)$ with probability zero.

By assumption the agent's policy is deterministic, and $\max\{P_n(X \leq k), P_n(X > k)\} > 0$, so for any $n, k$ the agent must take action $A = a$ or $A = b$ at $t = 0$. Therefore for any given $n, k$ the agent's policy $\pi(a_0 \mid s_0; \psi_{a,b}(k, n))$ selects either $a_0 = a$ or $a_0 = b$, and the choice of $A_0$ for a given $k$ witnesses the following inequalities;

$$\pi(a_0 \mid s_0; \psi_{a,b}(k, n)) = I([a_0 = a]) \implies P_n(X \leq k) \geq P_n(X > k)(1 - \delta) \tag{39}$$

$$\pi(a_0 \mid s_0; \psi_{a,b}(k, n)) = I([a_0 = b]) \implies P_n(X > k) \geq P_n(X \leq k)(1 - \delta) \tag{40}$$

For ease of notation we denote $P_{ss'}(a) = p$. The median of the binomial distribution $X = m$ is an integer $0 \leq m \leq n$ that satisfies $np - 1 \leq m \leq np + 1$. The proof proceeds by incrementing $k$ from 0 to $n$, increasing $P_n(X \leq k)$ while decreasing $P_n(X > k)$, and finding the smallest value $k^*$ such that $P_n(X > k^* - 1) \geq P_n(X \leq k^* - 1)(1 - \delta)$ and $P_n(X \leq k^*) \geq P_n(X > k^*)(1 - \delta)$. If the agent always chooses $A_0 = a$ we set $k^* = 0$, and if they always choose $A_0 = b$ we set $k^* = n$. This will turn out to be equivalent to a *sparsity bias* in the procedure for estimating $P_{ss'}(a)$, as it will result in us treating any transition probability below a given threshold value as 0, or above a maximum value as 1. Note that $P_n(X > 0) = 1, P_n(X \leq 0) = 0$, and $P_n(X > n) = 0, P_n(X \leq n) = 1$, so for any $\delta < 1$ there must

exist $0 \leq k^* \leq n$ satisfying (39) and (40). Using $P_n(X > k) = 1 - P_n(X \leq k)$, $P_n(X > k - 1) = P_n(X \geq k)$ and $P_n(X \leq k - 1) = 1 - P_n(X \geq k)$ these inequalities simplify to,

$$P(X \leq k^*) \geq \frac{1 - \delta}{2 - \delta} \tag{41}$$

$$P(X \geq k^*) \geq \frac{1 - \delta}{2 - \delta} \tag{42}$$

Note that the median $m$ satisfies $P_n(X \geq m) \geq 1/2$ and $P_n(X \leq m) \geq 1/2$, so for $\delta = 0$ we will recover the median exactly, and $\delta > 0$ constitutes a relaxation of the bounds on the median, which we will show results in $k^*$ that has a bounded distance from the mean $np$.

We derive two bounds on $k^*$, first in the case where $\delta$ is small and $n$ is large. To derive this first bound we use Berry-Esseen theorem, which allows us to bound the distance of the (normalised) cumulative binomial distribution from the cumulative normal distribution,

$$\left| P_n \left( \frac{X - np}{\sqrt{np(1 - p)}} \leq k \right) - \Phi(X \leq k) \right| \leq \Delta \tag{43}$$

where $\Phi$ is the cumulative normal distribution and $\Delta = C\rho/\sqrt{n}$ where $C$ is a constant satisfying $C \leq 0.4748$ and $\rho = (1 - 2p(1 - p))/\sqrt{p(1 - p)}$. For simplicity we relax this upper bound by taking,

$$\Delta = \frac{1}{2\sqrt{np(1 - p)}} \tag{44}$$

which is larger than $C\rho/\sqrt{n}$. Defining $Y := (X - np)/\sqrt{np(1 - p)}$, and using $P_n(Y \geq k) = 1 - P(Y \leq k - 1)$, (42) and (41) become,

$$\Phi \left( Y \leq \frac{k^* - np}{\sqrt{np(1 - p)}} \right) \geq \frac{1 - \delta}{2 - \delta} - \Delta \tag{45}$$

$$\Phi \left( Y \leq \frac{k^* - np - 1}{\sqrt{np(1 - p)}} \right) \leq \frac{1}{2 - \delta} + \Delta \tag{46}$$

$$\tag{47}$$

which can be rearranged to give,

$$\frac{k^* - np}{\sqrt{np(1 - p)}} \geq \Phi^{-1} \left( \frac{1 - \delta}{2 - \delta} - \Delta \right) \tag{48}$$

$$\frac{k^* - np - 1}{\sqrt{np(1 - p)}} \leq \Phi^{-1} \left( \frac{1}{2 - \delta} + \Delta \right) \tag{49}$$

$$\tag{50}$$

For $\delta \ll 1$ and $\Delta \ll 1$ we can approximate the right hand side of (48) and (49) using the Taylor expansion of $\Phi^{-1}(y)$ at $y = 1/2$,

$$\Phi^{-1}(Y = \frac{1}{2} + \epsilon) = \epsilon\sqrt{2\pi} + \mathcal{O}(\epsilon^3), \quad \epsilon \ll 1 \tag{51}$$

which is a valid approximation when $\epsilon^2 \ll 1$. We therefore recover the bounds,

$$k^* - \frac{1}{2} - np \gtrsim -\sqrt{2\pi np(1 - p)} \left( \frac{\delta}{4} + \Delta \right) - \frac{1}{2} \tag{52}$$

$$k^* - \frac{1}{2} - np \lesssim \sqrt{2\pi np(1 - p)} \left( \frac{\delta}{4} + \Delta \right) + \frac{1}{2} \tag{53}$$

Using $\hat{p} = (k^* - 1/2)/n$ as our estimate of $p$ therefore satisfies

$$|\hat{p} - p| \lesssim \sqrt{\frac{2\pi p(1-p)}{n}} \left(\frac{\delta}{4} + \Delta\right) + \frac{1}{2n}$$

$$= \delta\sqrt{\frac{\pi p(1-p)}{8n}} + \frac{1}{n}\left(\frac{1}{2} + \sqrt{2\pi}\right)$$

which is valid for $\delta^2 \ll 1$ and $\Delta^2 = 1/(np(1-p)) \ll 1$ i.e. $np(1-p) \gg 1$. We have therefore shown that in this regime the approximation errors scales as $\mathcal{O}(\delta/\sqrt{n}) + \mathcal{O}(1/n)$.

Finally, we derive an absolute error bound for the estimate $\hat{p}$ that is valid for all values of $p, n, \delta$. I.e. for $\delta \not\ll 1$ and/or $np(1-p) \not\gg 1$, $k^*$ can be relatively far from the median, and so we require a bound that is satisfied for the tails of the cumulative binomial distribution. To this end we apply the one-sided Chebyshev inequality,

$$P_n(X \geq \mu + t\sigma) \leq \frac{1}{1+t^2} \tag{54}$$

where $\mu = np$ and $\sigma = \sqrt{np(1-p)}$. Changing variables $t = (k^* - np)/\sigma$ in (54) yields,

$$P(X \geq k^*) \leq \frac{1}{1 + \frac{(k^* - np)^2}{np(1-p)}} \tag{55}$$

which combined with (42) yields,

$$|k^* - np| \leq \sqrt{\frac{np(1-p)}{1-\delta}} \tag{56}$$

Using $\hat{p} := k^*/n$ as our estimate of the transition probability gives,

$$|\hat{p} - p| \leq \sqrt{\frac{p(1-p)}{n(1-\delta)}} \tag{57}$$

Finally, as attempting the transition $n$ times corresponds to a goal depth of $2n + 1$, we arrive at our final expression.

$\square$

Note that the bound in Theorem 1 asserts that we can identify $p \in \{0, 1\}$ with perfect precision. While this appears surprising at first, note that the sparsity constraint we previously enforced when selecting $k^*$ means that we estimate all sufficiently small $p$ as $\hat{P}_{ss'}(a) = 0$ and similar for $\hat{P}_{ss'}(a) = 1$, hence for deterministic transition probabilities we do recover the exact value. This is also intuitive from the definition of the bounded goal-conditioned agent Def. 5, as for any $\delta < 1$ the agent will never choose sub-goal $\psi_a$ if $P_{ss'}(a) = 0$, and hence any such transition will always be assigned $k^* = 0$ which yields an estimate $\hat{P}_{ss'}(a) = 0$.

## B. Proof of Theorem 2

**Theorem 2.** *Let the set of myopic goals $\Psi_{myopic}$ be the subset of depth-1 composite goals $\Psi_1$ such that the goal state(s) must be attained immediately after the agents first action, $\varphi = \bigcirc[(s, a) \in \boldsymbol{g}]$. We define an optimal myopic agent as a policy $\pi^*(a_t \mid h_t, \psi)$ that is optimal for all $\psi \in \Psi_{myopic}$. For an environment satisfying Assumption 1, any bounds on the transition probabilities $|\hat{P}_{ss'}(a) - P_{ss'}(a)| \leq \epsilon$ than can be determined from $\pi^*$ are trivial ($\epsilon = 1$) and tight. Proof in Appendix B.*

*Proof.* We will prove this by contradiction, determining partial information of the environment transition function that is sufficient to construct an optimal myopic agent, and showing that this partial information is insufficient to bound the transition probabilities (the trivial bound is tight). Hence, there can exist no procedure that bounds the transition probabilities given this partial information, and so no procedure that does so given the optimal myopic policy.

Any $\psi \in \Psi_{\text{myopic}}$ is of the form $\varphi_1 \vee \varphi_1 \vee \ldots \vee \varphi_k$ where $\varphi_i = \bigcirc([(s, a) \in \boldsymbol{g}_i)$. Using the transitivity of the Next operator, this can be simplified to $\psi = \bigcirc([(s, a) \in \boldsymbol{g}_1] \vee \ldots \vee [(s, a) \in \boldsymbol{g}_k])$, and using $\boldsymbol{y} = \boldsymbol{g}_1 \cup \boldsymbol{g}_2 \cup \ldots \cup \boldsymbol{g}_k$ we get

$\psi = [(s_1, a_1) \in \boldsymbol{y}]$ where $\boldsymbol{Y} \subseteq \boldsymbol{S}$ is some arbitrary subset of $\boldsymbol{S}$. For $A_0 = a$ the probability that $\psi$ is satisfied is given by $P(\tau \models \psi \mid \pi, s_0) = P(s_1 \in \boldsymbol{y} \mid a, s_0)$. The optimal agent therefore returns an action

$$\pi^*(a_0 \mid s_0; \psi) = \arg\max_a P(s_1 \in \boldsymbol{y} \mid a, s_0) \tag{58}$$

Let $a^*(s_0, \boldsymbol{y}) := \arg\max_a P(s_1 \in \boldsymbol{y} \mid a, s_0)$. We can construct an optimal policy $\pi^*(a_0 \mid s_0; \psi)$ given $\boldsymbol{A}^* = \{a^*(s_0, \boldsymbol{y}) \mid s_0, \boldsymbol{Y} \subseteq \boldsymbol{S}, \boldsymbol{Y} = \boldsymbol{y}\}$ as $\pi^*(a_0 \mid s_0; \psi) = I([a_0 = a^*(s_0, \boldsymbol{y})])$. Next, we show that the set of transition functions that are compatible with any given $\boldsymbol{A}^*$ includes all values of $P_{ss'}(a) \in [0, 1]$ for any given transition, and so $\boldsymbol{A}^*$ does not partially identify $P_{ss'}(a)$. This can be seen simply by choosing $P_{ss'}(a) = P_{ss'}$ (i.e. the transition probabilities are the same for all actions). For any choice of $P_{ss'} \in [0, 1]$, such a transition function is compatible with all possible $\boldsymbol{A}^*$. Hence, for any given $\boldsymbol{A}^*$ the set of compatible values of $P_{ss'}(a)$ is $[0, 1]$, and knowing $\boldsymbol{A}^*$ provides no bound on the possible values of any given transition probability $P_{ss'}(a)$ (i.e. partial identification is impossible (Bellot, 2023)). Hence, as $\pi^*$ is a function of $\boldsymbol{A}^*$, $\pi^*$ can provide no non-trivial bound on $P_{ss'}(a)$.

$\square$

## C. Algorithms

First we present the pseudocode for the procedure Algorithm 1 used in the proof of Theorem 1 to derive error-bounded estimates of the transition probabilities $\hat{P}_{ss'}(a)$ given the regret-bounded goal-conditioned policy $\pi(a_t \mid h_t; \psi)$. We then present Algorithm 2 an alternative algorithm for estimating $\hat{P}_{ss'}(a)$ which has weaker errors bounds than Algorithm 1 but significantly simplified implementation. Note that in both Algorithm 1 and Algorithm 2 we employ a linear search of $k$, but we can greatly reduce the complexity in practice e.g. by performing a binary search over $k \in [0, n]$. We use Algorithm 2 to generate our experimental results in Section 3.1 and Appendix D. Note that by looping over all transitions $(s, a, s')$ and applying Algorithm 1 we can recover the full transition function.

---

**Algorithm 1** Estimate Transition Probability $\hat{P}_{ss'}(a)$ from Policy $\pi$

---

**Require:** Goal-conditioned policy $\pi(a_t|h_t; \psi)$

**Require:** Choice of state $s$, action $a$, outcome $s'$

**Require:** Precision parameter $n \in \mathbb{N}$ (related to maximum goal depth $2n + 1$)

**Require:** An alternative action $b \neq a$
1: **function** ESTIMATETRANSITIONPROBABILITY$(\pi, s, a, s', n, b)$
2:      Initialize $k^* \leftarrow n$
3:      **for** $k = 1$ to $n$ **do**
4:          Define base LTL components:
5:             $\varphi_0 \leftarrow [A_0 = a]$
6:          $\triangleright$      *Take action $a$*          $\triangleleft$
7:             $\varphi'_0 \leftarrow [A_0 = b]$
8:          $\triangleright$      *Take action $b$*          $\triangleleft$
9:             $\varphi_1 \leftarrow \Diamond[A = a, S = s]$
10:     $\triangleright$      *Transitions eventually to state $s$ and takes action $a$*          $\triangleleft$
11:             $\varphi_2 \leftarrow \bigcirc[S = s']$
12:     $\triangleright$      *Transition Next to state $s'$*          $\triangleleft$
13:             $\varphi'_2 \leftarrow \bigcirc[S \neq s']$
14:     $\triangleright$      *Transition Next to any state other than $s'$*          $\triangleleft$
15:          Define composite goal:
16:          $\psi_0 \leftarrow \langle \varphi_1, \varphi'_2 \rangle$
17:     $\triangleright$      *Sequential goal labelled Fail*          $\triangleleft$
18:          $\psi_1 \leftarrow \langle \varphi_1, \varphi_2 \rangle$
19:     $\triangleright$      *Sequential goal labelled Success*          $\triangleleft$
20:          $\psi_a(k, n) \leftarrow \bigvee_{\text{sequences with } r \leq k \text{ successes}} \langle \varphi_0, (\psi_0 \text{ or } \psi_1)_{\times n} \rangle$
21:          $\psi_b(k, n) \leftarrow \bigvee_{\text{sequences with } r > k \text{ successes}} \langle \varphi'_0, (\psi_0 \text{ or } \psi_1)_{\times n} \rangle$
22:     $\triangleright$      *LTL expressions calculated with Def. 6*          $\triangleleft$
23:          $\psi_{a,b}(k, n) \leftarrow \psi_a(k, n) \vee \psi_b(k, n)$
24:          $a_0 \leftarrow \pi(a_0|s_0; \psi_{a,b}(k, n))$
25:     $\triangleright$      *Query the policy for the first action*          $\triangleleft$
26:          **if** $a_0 = a$ **then**
27:             $k^* \leftarrow k$
28:             **break**
29:         $\triangleright$      *Found smallest $k$ s.t. where agent prefers goal involving $\leq k$ successes*          $\triangleleft$
30:      Estimate $\hat{P}_{ss'}(a) \leftarrow (k^* - 1/2)/n$
31:      **return** $\hat{P}_{ss'}(a)$

---

Algorithm 2 requires the agent to generalize to simpler sequential goals than Algorithm 1.

---

**Algorithm 2** Simplified method for estimating Transition Probability $\hat{P}_{ss'}(a)$ from Policy $\pi$ with weaker error bounds than Algorithm 1

---

**Require:** Goal-conditioned policy $\pi(a_t|h_t; \psi)$

**Require:** Choice of state $s$, action $a$, outcome $s'$

**Require:** Precision parameter $n \in \mathbb{N}$ (related to maximum goal depth $2n + 1$)

**Require:** An alternative action $b \neq a$

1:   **function** ESTIMATETRANSITIONPROBABILITY($\pi, s, a, s', n, b$)
2:     Define base LTL components:
3:     $\varphi_0 \leftarrow [A_0 = a]$
4:     $\triangleright$    *Take action $a$*                                   $\triangleleft$
5:     $\varphi_0' \leftarrow [A_0 = b]$
6:     $\triangleright$    *Take action $b$*                                   $\triangleleft$
7:     $\varphi_1 \leftarrow \Diamond[A = a, S = s]$
8:     $\triangleright$    *Transitions eventually to state $s$ and takes action $a$*    $\triangleleft$
9:     $\varphi_2 \leftarrow \bigcirc[S = s']$
10:    $\triangleright$    *Transition Next to state $s'$*                     $\triangleleft$
11:    $\varphi_2' \leftarrow \bigcirc[S \neq s']$
12:    $\triangleright$    *Transition Next to any state other than $s'$*      $\triangleleft$
13:    Define sequential goals:
14:    $\psi_a \leftarrow \langle \varphi_0, \varphi_1, \varphi_2 \rangle$
15:    $\psi_b \leftarrow \langle \varphi_0, \varphi_1, \varphi_2' \rangle$
16:    $\psi_{a,b} = \psi_a \vee \psi_b$
17:    $a_0 \leftarrow \pi(a_0|s_0; \psi_{a,b})$
18:    $\triangleright$    *Query the policy for the first action*            $\triangleleft$
19:    **if** $a_0 = a$ **then**
20:      $\triangleright$     *Witnessing $P_{ss'}(a) \geq (1 - P_{ss'}(a))(1 - \delta)$*    $\triangleleft$
21:      $\psi_a \leftarrow \langle \varphi_0, (\psi_1, \psi_2)_{\times n} \rangle$
22:      $\psi_b(k) \leftarrow \langle \varphi_0, (\psi_1, \psi_2')_{\times k} \rangle$
23:      **for** $k = 1$ to $n$ **do**
24:        $\psi_{a,b}(k) \leftarrow \psi_a \vee \psi_b(k)$
25:        $a_0 \leftarrow \pi(a_0|s_0; \psi_{a,b}(k))$
26:        $\triangleright$    *Query the policy for the first action*     $\triangleleft$
27:        **if** $a_0 = a$ **then**
28:          $k^* \leftarrow k$
29:          **break**
30:      Estimate $\hat{P}_{ss'}(a) \leftarrow \text{Solve}(P^n = (1 - P)^{k^* - 1/2})$
31:      **return** $\hat{P}_{ss'}(a)$
32:    **else**
33:      $\psi_b \leftarrow \langle \varphi_0, (\psi_1, \psi_2')_{\times n} \rangle$
34:      $\psi_a(k) \leftarrow \langle \varphi_0, (\psi_1, \psi_2)_{\times k} \rangle$
35:      **for** $k = 1$ to $n$ **do**
36:        $\psi_{a,b}(k) \leftarrow \psi_a(k) \vee \psi_b$
37:        $a_0 \leftarrow \pi(a_0|s_0; \psi_{a,b}(k))$
38:        $\triangleright$    *Query the policy for the first action*     $\triangleleft$
39:        **if** $a_0 = b$ **then**
40:          $k^* \leftarrow k$
41:          **break**
42:      Estimate $\hat{P}_{ss'}(a) \leftarrow \text{Solve}(P^{k^* - 1/2} = (1 - P)^n)$
43:      **return** $\hat{P}_{ss'}(a)$

---

## D. Experiments

Here we detail the experiment setup including the environment, agent and results.

**Environment.** Our environment is a cMP Def. 1 comprising of 20 states and 5 actions, and satisfying Assumption 1. It has a randomly generated transition function with a sparsity constraint such that each state-action pair has at most 5 outcomes that occur with non-zero probability, so as to ensure that navigating eventually to a given goal-state is non-trivial (e.g. is not achieved by all deterministic policies).

**Agent.** The agent is model based, with the model learned from experienced generated by sampling state-action trajectories from the environment under the maximally random policy of a given number of time steps $N_{\text{samples}} \in \{500, 1000, 2000, 3000, 4000, 5000, 6000, 7000, 8000, 9000, 10,000\}$. Note that Algorithm 2 does not have access to the agents internal world model (the algorithm takes as input only the agent's policy). Algorithm 2 queries the agent with different composite goals of the form $\psi_{a,b}(n, m)$, and the agent determines the optimal policy with respect to its world model, which corresponds to 1) at $t = 0$ taking action $A = a$ if the agent believes $P_{ss'}(a)^n > (1 - P_{ss'}(a))^m$ else $A = b$ 2) identifying a deterministic policy that eventually reaches the target state $S = s$ from any other state, and taking action $A = a$ in $S = s$.

**Experimental setup.** We train 10 agents for each sample size $N_{\text{samples}}$, with a different random seed for the experience trajectories, and take the average of the experimental results over the set of agents with the same sample size. For each agent we run Algorithm 2 for different max goal depths $N \in \{10, 20, 50, 75, 100, 200, 300, 400, 500, 600\}$, and record the regret $\delta$ for each input goal, which is $1 - P(\tau \models \psi_{n,m} \mid \pi)/P(\tau \models \psi_{n,m} \mid \pi^*)$ where $P(\tau \models \psi_{n,m} \mid \pi)$ is the probability the agent achieves the goal agent's policy and $P(\tau \models \psi_{n,m} \mid \pi^*)$ is the probability that the optimal policy achieves the goal. We then calculate the average regret $\langle \delta \rangle$ all goals the agent is queried with by Algorithm 2, and the average error $\langle \epsilon \rangle$ (averaged over all state-action-outcome tuples) for the estimated transition function returned by Algorithm 2. We determine $N_{\text{max}}(\langle \delta \rangle = k)$ through least-squares regression of $N$ (goal depth) v.s. $\langle \delta \rangle$ for a given agent.

**Results.**

Table 1: Mean Error and Standard Deviation for Different $N_{\text{samples}}$ and $N_{\text{depth}}$ Values

| $N_{\text{depth}}$ | $N_{\text{samples}}$ | | | | | | | | | | |
|---|---|---|---|---|---|---|---|---|---|---|---|
| | 500 | 1000 | 2000 | 3000 | 4000 | 5000 | 6000 | 7000 | 8000 | 9000 | 10000 |
| 10 | 0.171 ± 0.007 | 0.137 ± 0.008 | 0.111 ± 0.009 | 0.097 ± 0.006 | 0.088 ± 0.005 | 0.082 ± 0.003 | 0.078 ± 0.003 | 0.076 ± 0.004 | 0.075 ± 0.004 | 0.072 ± 0.005 | 0.066 ± 0.005 |
| 20 | 0.160 ± 0.008 | 0.118 ± 0.008 | 0.088 ± 0.005 | 0.073 ± 0.004 | 0.064 ± 0.002 | 0.059 ± 0.003 | 0.054 ± 0.003 | 0.052 ± 0.003 | 0.049 ± 0.003 | 0.047 ± 0.002 | 0.044 ± 0.003 |
| 50 | 0.157 ± 0.008 | 0.108 ± 0.008 | 0.077 ± 0.005 | 0.063 ± 0.003 | 0.054 ± 0.003 | 0.048 ± 0.003 | 0.044 ± 0.003 | 0.041 ± 0.003 | 0.039 ± 0.003 | 0.037 ± 0.002 | 0.034 ± 0.002 |
| 75 | 0.157 ± 0.008 | 0.107 ± 0.008 | 0.075 ± 0.005 | 0.061 ± 0.003 | 0.052 ± 0.003 | 0.047 ± 0.002 | 0.042 ± 0.002 | 0.040 ± 0.003 | 0.038 ± 0.003 | 0.035 ± 0.002 | 0.033 ± 0.002 |
| 100 | 0.156 ± 0.008 | 0.106 ± 0.008 | 0.074 ± 0.004 | 0.060 ± 0.003 | 0.051 ± 0.002 | 0.046 ± 0.002 | 0.041 ± 0.002 | 0.039 ± 0.003 | 0.037 ± 0.003 | 0.034 ± 0.002 | 0.032 ± 0.002 |
| 200 | 0.155 ± 0.008 | 0.105 ± 0.007 | 0.073 ± 0.004 | 0.059 ± 0.003 | 0.050 ± 0.002 | 0.045 ± 0.002 | 0.040 ± 0.002 | 0.038 ± 0.003 | 0.036 ± 0.003 | 0.034 ± 0.002 | 0.031 ± 0.002 |
| 300 | 0.155 ± 0.008 | 0.104 ± 0.007 | 0.072 ± 0.004 | 0.058 ± 0.003 | 0.049 ± 0.002 | 0.044 ± 0.002 | 0.040 ± 0.002 | 0.038 ± 0.003 | 0.036 ± 0.003 | 0.033 ± 0.002 | 0.031 ± 0.002 |
| 400 | 0.155 ± 0.008 | 0.104 ± 0.007 | 0.072 ± 0.004 | 0.058 ± 0.003 | 0.049 ± 0.002 | 0.044 ± 0.002 | 0.040 ± 0.002 | 0.038 ± 0.003 | 0.035 ± 0.003 | 0.033 ± 0.002 | 0.031 ± 0.002 |
| 500 | 0.155 ± 0.008 | 0.104 ± 0.007 | 0.072 ± 0.004 | 0.058 ± 0.003 | 0.049 ± 0.002 | 0.044 ± 0.002 | 0.040 ± 0.002 | 0.037 ± 0.003 | 0.035 ± 0.003 | 0.033 ± 0.002 | 0.031 ± 0.002 |
| 600 | 0.155 ± 0.008 | 0.104 ± 0.007 | 0.072 ± 0.004 | 0.058 ± 0.003 | 0.049 ± 0.002 | 0.044 ± 0.002 | 0.040 ± 0.002 | 0.037 ± 0.003 | 0.035 ± 0.003 | 0.034 ± 0.002 | 0.031 ± 0.002 |

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
