# OpenReview forum: "General agents need world models"
_ICML.cc/2025/Conference — ICML 2025 poster_

### Official Review · Reviewer_idip · 2025-03-10

**Overall Recommendation:** 3

**Summary:**

- This paper proves a bound on an agent's ability to achieve zero-shot generalization.
- It studies a full-observable controlled Markov process, with standard simplifying environment assumptions.
- They find a bound on the regret of an agent with a key term being an L1 distance between true and estimated transition probabilities.
- This leads the paper to conclude that to do zero-shot generalization, agents must learn world models.
- There is lengthy discussion on implications of this and how it connects to results in related areas such as causality and safety.

**Claims And Evidence:**

- While I don't disagree with the discovered bound, I'm unclear whether this by itself definitively supports some claims such as 'Theorem 1 shows that any bounded agent has learned a world model'. The implicit vs explicit distinction seems important detail to maintain in such claims.

- Another key contribution of the paper is 'we can recover an approximation of the environment transition function (a world model) from the agents policy alone', but I found the explanation of how to do this (at least in the main paper) vague.

**Essential References Not Discussed:**

Fine.

**Experimental Designs Or Analyses:**

No experiments.

**Methods And Evaluation Criteria:**

No experiments.

**Other Comments Or Suggestions:**

- In abstract, why is it specifically a _generative_ model that is learned?
- 'three temporal operators', but seemed to only define two (is the later or operation included in this?)
- Should eq 2 on rhs be $\pi^*$?
- What is $p$ in theorem 1? did I miss something?
- 'While model-based agents explicitly learn world models (typically transformers (Brooks et al., 2024) or diffusion models (Janner et al., 2022))'. Why cite Brooks et al. here? They do not do model-based learning, they are doing video generation. (They also use diffusion.)
- Line 206 rh column, I don't see what this prob ratio refers to?
- The first few pages of the appendix repeat most of the main paper again. If there are important details in the appendix, move them to the main paper.

**Other Strengths And Weaknesses:**

- From one angle, the paper seems to be saying something quite obvious in a very complicated way -- that an agent must have an internal estimate of how its environment works in order to be good at reaching states within it. The better its internal estimates, the better the agent. I'm not familiar with the thought history on this, and I get lost in some of the nuances of the discussion, but to me this seems quite intuitive. From this angle I'm unclear how much impact the paper's result has.
- I'm unfamiliar with the structure the paper uses -- lengthy set up, followed by a very short summary of the main theoretical result, and then long discussion of connections with other work. No experiments.
- The paper takes just 7 pages, including a fair bit of repetition of text and definitions. Overall it feels a little empty. The heart of the contribution is section 3, which takes up less than a page.
- The paper could use this spare space to do a much better job of giving insight into theorem 1's proof and the procedure for extracting the transition probabilities.

**Questions For Authors:**

See review.

**Relation To Broader Scientific Literature:**

I am not particularly familiar with related work so withhold judgement.

**Theoretical Claims:**

Assessing these proofs is a little out my comfort area. I can take a deeper dive if needed depending on other reviewer's expertise. For the first pass, I read the main paper carefully but only spot checked the proofs in the appendix.

So far the technical details appear well written and correct.

---

> ### Author Rebuttal · Authors · 2025-03-26
>
> Thank you for your detailed and thoughtful review and helpful comments. We hope to address your main concerns about the core claims of our paper, which we believe stem from a misunderstanding of theorem 1, and have implemented your suggestions for improving the paper.
>
> **What do we actually show?**
> Your review notes that we _prove a bound on the agent's ability to achieve zero-shot generalization_ and from this conclude that to do zero-shot generalization, agents must learn world models. We agree that if this was our result, it would be highly questionable.
>
> What we actually do is _assume_ the agent satisfies this regret bound (rather than prove it does). I.e. we assume the agent has some minimal degree of competence at zero-shot learning. We then formally prove (as verified by reviewer Xhe1) that for any agent satisfying this assumption, a world model is encoded in the agent's policy. We derive an (unsupervised) algorithm for recovering this world model from the policy (similar to [4]), and prove an error bound on the accuracy of the world model recovered, which depends on the agent's regret.
>
> We agree that this could be made clearer in the paper, and have done so by specifying the algorithm that recovers the world model outside of the proof of Theorem 1 (see Algorithm 1 in response to reviewer aY71) and re-writing the results section to improve clarity and discussion of our results.
>
> **Is this obvious?**
> The question of whether or not AI systems have or need world models is hotly debated (see for example [1]), and the subject of significant empirical research [2,3]. A similar result to ours [4] was recently proven for domain generalization (rather than zero-shot learning) and received an award at ICLR 2024.
>
> While we agree it feels intuitive that agents should have a world model, there are many ways agents can can reach goal states without one; e.g. through numerous heuristics (schemas, similarity-based reasoning, ...), online learning, etc. Humans can switch between using model-based reasoning _or_ heuristics to generalize to new tasks, depending on the situation. And many biological agents that are thought to be purely model-free (`stimulus-response' agents [5]). It is unclear if general AI systems like LLMs have world models, or if they can generalize purely via heuristics. Before now, there was no formal result showing that world models are necessary for generalization. And indeed we show that for myopic tasks, where the agent is optimizing for immediate outcomes, world models are _not_ necessary.
>
> We tackle this question by tying world models to a key capability---zero-shot learning. This has consequences for how we design agents (model-based v.s. model free) and reveals fundamental limitations on agent capabilities. Further, we show that this world model can be extracted, which has consequences for safety and interpretability.
>
> **How have we improved the paper based on your feedback**
>
> 1. We agree with your point that the paper lacks insights into theorem 1 and the procedure for extracting world models. We now give an explicit algorithm for recovering world models (see response to reviewer aY71), and have extended the discussion of theorem 1 and other results to 2 pages and reduced repetition.
>
> 2. We have included experiments demonstrating that our algorithm can be applied to real world agents (see response to reviewer aY71)
>
> 3. We have introduced a new theorem which proves that learning a world model is _not necessary_ for myopic agents, which optimize for immediate outcomes to their actions (depth-1 goals). This relates to your question on if the need for a world model is obvious or not. Theorem and proof can be seen here https://imgur.com/a/QzrXt0W
>
>
>
> **Reviewer questions**
>
> 1. _Why is it a generative model?_. The world model we recover can be used to simulate environment trajectories. This is opposed to the purely state-based world models often studied (e.g. in [2]).
> 2. _Three temporal operators..._ Typo corrected. We were referring to the trivial (Now) operator, but have removed this.
> 3. _Should eq. 2 rhs be $\pi^*$_ We are taking the max over $\pi$, which is equivalent.
> 4. _What is $p$ in theorem 1_. Typo corrected.
> 5. Comments on Brookes et. al. 2024, have removed.
> 6. _Line 206 rh column, I don't see what this prob ratio refers to?_. This is the ratio of the estimated transition probability to the true value, i.e. the relative error. It is given by dividing the inequality in theorem 1 by $P_{ss'}(a)$
>
>
>
>
> [1] https://x.com/ylecun/status/1667947166764023808
>
> [2] Li, et al. "Emergent world representations: Exploring a sequence model trained on a synthetic task." ICLR (2023) oral
>
> [3] Gurnee et. al. "Language Models Represent Space and Time." ICLR (2024)
>
> [4] Richens et. al. "Robust agents learn causal world models". ICLR (2024) oral
>
> [5] Tomasello. The evolution of agency: Behavioral organization from lizards to humans. MIT Press, 2022.

---

> > ### Comment · Reviewer_idip · 2025-04-03
> >
> > Thank you for this rebuttal. I have nudged my score up slightly to reflect my misunderstanding of the proof.

---

### Official Review · Reviewer_aY71 · 2025-03-10

**Overall Recommendation:** 3

**Summary:**

This paper shows the insight that any agent capable of performing zero-shot generalization must have learned an accurate generative model as a world model of its environment. This paper provides a comprehensive theoretical analysis to support the claims.

## update after rebuttal
Thanks to the authors for providing the rebuttal. I've read the author's response and comments from other reviewers. I have no further questions at this time. I will keep my original positive rating.

**Claims And Evidence:**

This paper provides detailed theoretical proof to stand for the claims.

**Essential References Not Discussed:**

To the best of my knowledge, the references are sufficiently covered.

**Experimental Designs Or Analyses:**

This paper does not provide any experiment.

**Methods And Evaluation Criteria:**

This paper only provides a theoretical framework but does not propose a new method and does not conduct experiments.

**Other Comments Or Suggestions:**

Please refer to the issues raised above.

**Other Strengths And Weaknesses:**

While this paper focuses on the theoretical part, some experiments, even in some simple environments like Atari, help readers connect the claims of this paper to real-world RL or robotics applications. Whether any tasks exist (e.g., robot navigation, manipulation) can this paper's claims be applied?

**Questions For Authors:**

Please refer to the issues raised above.

**Relation To Broader Scientific Literature:**

This paper provides a theoretical framework to claim that the world model is essential for zero-shot generalization.

**Theoretical Claims:**

I have checked the main theoretical claims, but not in every detail.

---

> ### Author Rebuttal · Authors · 2025-03-26
>
> Thank you for you helpful comments. As noted by **Reviewer Xhe1**, our paper does propose a new method for eliciting world models from agents. However, this was quite unclear in the submitted draft, and we have included an explicit algorithm (below) in the manuscript to clarify this.
>
> Following you recommendation we have made the following changes to the paper;
>
> 1. Explicit algorithms for recovering world models from agents (below)
> 2. New experiments, validating these algorithms on real agents (details below).
> 3. Discussion of real-world tasks where our results can be applied (for example [1] recently developed goal conditioned agents that can generalize zero-shot to arbitrary linear temporal logic goals).
>
>
> [1] Jackermeier et al. "DeepLTL: Learning to Efficiently Satisfy Complex LTL Specifications for Multi-Task RL." ICLR 2025 (oral)
>
>
> ### **New experiments**
>
> Motivation: can our algorithm for recovering an agent's world model (below) be applied to real-world agents that perhaps maximally violate our theoretical assumptions? Namely, the strict regret bound we assume in Theorem 1.
>
> **Experimental setup**
>
> Our experiment involves extracting a world model from a model-based language agent in 120 randomly generated cMDP environments using our algorithm (below). We show the agent strongly violates our assumptions, but nonetheless this algorithm can still recover the agent's world model.
>
> 1. 120 randomly generated environments described by cMPs, with between 5 and 40 states and 3 and 20 actions.
> 2. Our goal-conditioned agent is an LLM (Gemini Flash 2.0), with an explicit, private world model.
> 3. We then attempt to learn this private world model using Algorithm 1 (above) given only the agent's policy
>
> A figure of our results can be viewed at the following URL: https://imgur.com/a/1gNe15c
>
> We also note that our paper is a theory paper, which as pointed out by reviewer Xhe1 significantly extends important recent theory work. We hope it can be judged on these merits, without requiring experiments that extend on the current state of the art empirical work (for example [1] trained LTL conditioned agents and was an oral at ICLR 2025, and used environments simpler than Atari).
>
> ### **Algorithm 1: Estimate Transition Probability $\hat{P}_{ss'}(a)$ from Policy $\pi$**
>
> **Input:**
> * Goal-conditioned policy $\pi(a_t | h_t; \psi)$
> * Choice of state $s$, action $a$, outcome $s'$
> * Precision parameter $n \in \mathbb{N}$ (related to maximum goal depth $2n+1$)
> * An alternative action $b \neq a$
>
> **Function:** EstimateTransitionProbability($\pi, s, a, s', n, b$)
>
> 1.  Initialize $k^* \gets n$
> 2.  For $k = 1$ to $n$:
>     * Define base LTL components:
>         * $\varphi_0 \gets [A_0=a]$  (*Take action $a$*)
>         * $\varphi'_0 \gets [A_0=b]$ (*Take action $b$*)
>         * $\varphi_1 \gets ◇ [A=a, S = s]$ (*Transitions eventually to state $s$ and takes action $a$*)
>         * $\varphi_2 \gets ○ [S=s']$ (*Transition Next to state $s'$*)
>         * $\varphi'_2 \gets ○ [S\neq s']$ (*Transition Next to any state other than $s'$*)
>     * Define composite goal:
>         * $\psi_0 \gets \langle\varphi_1, \varphi_2'\rangle$ (*Sequential goal labelled Fail*)
>         * $\psi_1 \gets \langle\varphi_1, \varphi_2\rangle$ (*Sequential goal labelled Success*)
>         * $\psi_a(k,n) \gets \bigvee_{\text{sequences with } r \le k \text{ successes}} \langle \varphi_0, (\psi_0 \text{ or } \psi_1)_{\times n} \rangle$
>         * $\psi_a(k,n) \gets \bigvee_{\text{sequences with } r > k \text{ successes}} \langle \varphi_0', (\psi_0 \text{ or } \psi_1)_{\times n} \rangle$
>     * $\psi_{a,b}(k,n) \gets \psi_a(k,n) \vee \psi_b(k,n)$
>     * $a_0 \gets \pi(a_0 | s_0; \psi_{a,b}(k,n))$ (*Query the policy for the first action*)
>     * If $a_0 = a$:
>         * $k^* \gets k$
>         * **break** (*Found smallest $k$ s.t. where agent prefers goal involving $\le k$ successes*)
> 3.  Estimate $\hat{P}_{ss'}(a) \gets (k^*-1/2)/n$
> 4.  **Return** $\hat{P}_{ss'}(a)$

---

> > ### Comment · Reviewer_aY71 · 2025-04-02
> >
> > Thanks to the authors for providing the rebuttal. I've read the author's response and comments from other reviewers. I have no further questions at this time. I will keep my original positive rating.

---

### Official Review · Reviewer_Xhe1 · 2025-03-11

**Overall Recommendation:** 4

**Summary:**

The authors establish that an agent capable of generalizing across a sufficiently large number of goal-conditioned tasks within an environment must have learned an accurate approximation of the environment’s transition model. As a consequence of this result, their proof provides a method for extracting the transition model directly from the agent's policy.

**Claims And Evidence:**

The main claim is supported by the proof of Theorem 1, which appears to be correct to the best of my knowledge.

However, the authors also claim that an agent trained on a small set of 'universal' goal-directed tasks can generalize to solve significantly more complex tasks. This claim raises concerns, as the proof requires the agent to successfully solve all composite goals up to a given maximum depth. It is unclear whether this set remains small relative to the set of all possible finite-time trajectories. Additionally, if the maximum depth constraint is reduced and model error increases, it is not evident how this would affect generalization performance beyond the training set.

**Essential References Not Discussed:**

N/A

**Experimental Designs Or Analyses:**

N/A

**Methods And Evaluation Criteria:**

N/A

**Other Comments Or Suggestions:**

p.5 second sentence "We compare two goals, the first ψ1(r, n) which is satisfied if the outcome state is S = s at most r times" -> Shouldn't the outcome state be s'?

**Other Strengths And Weaknesses:**

Strengths: The proofs are easy to read. Both the discussion and related work sections are insightful.
Weakness: It not obvious that the set of 'universal' goal-directed tasks is small compared to the set of finite trajectories. I would be happy to raise my score if the authors provide insights on this aspect.

**Questions For Authors:**

N/A

**Relation To Broader Scientific Literature:**

This paper extends the important findings of Richens (2024) to the sequential setting, albeit within a more restricted domain. Richens (2024) demonstrated that a robust agent must have learned a causal model, and this work builds upon that insight by considering goal-conditioned agents in sequential decision-making tasks. This line of research is particularly timely, as large goal-conditioned models are increasingly being deployed in real-world robotic applications.

**Theoretical Claims:**

I have carefully reviewed the proofs of the lemmas and the theorem, and to the best of my knowledge, they appear to be correct.

---

> ### Author Rebuttal · Authors · 2025-03-25
>
> Thank you for thoroughly reviewing our paper and for your helpful comments. In particular, the following comment on the need to clarify the size of the `universal’ set of goals, compared to the set of finite horizon trajectories.
>
> **Reviewer:** _It not obvious that the set of 'universal' goal-directed tasks is small compared to the set of finite trajectories. I would be happy to raise my score if the authors provide insights on this aspect._
>
> You correctly point out that we assume the agent can generalize to any composite goal of depth n (denoted $\mathbf{\Psi}_n$). What we failed to make clear in the paper is that the proof of Theorem 1 actually only requires that the agent can generalize to only a very small subset of $\mathbf{\Psi}_n$. This is what led us to comment on the existence of small `universal’ goal sets, which are sufficient for learning a world model and hence generalizing to more complex goals.
>
> Precisely, the set of goals we require the agent to generalize to is $\\{ \psi_{a, b}(k, n) \\}_{k=1}^{k=n}$
>
> where $\psi_{a, b}(k, n)$ are described at the start of the proofs of Lemma 5 and Theorem 1. The cardinality of this set is n, which is much smaller than $\mathbf{\Psi}_n$, and in general much smaller than the number of finite time trajectories, which scale as $\mathcal{O}(|S|^T |A|^{T-1})$ for horizon $T$. We can interpret this as growing exponentially with the goal depth n, as satisfying our sequential goals of depth n requires trajectories at least of length $T \geq n/2$ for these goals.
>
> We have made this all clear in the paper, and thank the reviewer for pointing it out.
>
> Typo corrected on p.5. second sentence.

---

### Decision · Program_Chairs · 2025-05-01

**Decision:**

Accept (poster)

**Comment:**

The paper establishes a theoretical result indicating that a reinforcement learning agent capable of generalising across a sufficiently large number of goal-conditioned tasks (where goals are LTL specifications) within an environment must necessarily have learned an accurate transition model of the environment.

The paper's result appears sound and is relevant to discussions on whether AI agents must or do learn "world models". The result here indicates that, at least in this context, learning a world model is a necessary condition if an agent is to be considered general. The paper is purely theoretical, establishing its main result and then discussing the implications; however, reviewers were split on whether the result is indeed valuable and useful to the RL community, so a better discussion here would improve the paper.  During the rebuttal, the authors provided clarifications, a practical algorithm for extracting the transition model and experimental results that align with the theory, although perhaps even better would have been a simpler experiment on a small cMP demonstrating how the error in the transition function changes as an agent has learned to achieve more goals.

Despite these shortcomings, the reviewers generally agree on acceptance, and authors are encouraged to incorporate the additional information provided during the rebuttal phase.